# Quantifying how single dose Ad26.COV2.S vaccine efficacy depends on Spike sequence features

In the ENSEMBLE randomized, placebo-controlled phase 3 trial (NCT04505722), estimated single-dose Ad26.COV2.S vaccine efficacy (VE) was 56% against moderate to severe–critical COVID-19. SARS-CoV-2 Spike sequences were determined from 484 vaccine and 1,067 placebo recipients who acquired COVID-19. In this set of prespecified analyses, we show that in Latin America, VE was significantly lower against Lambda vs. Reference and against Lambda vs. non-Lambda [family-wise error rate (FWER) p < 0.05]. VE differed by residue match vs. mismatch to the vaccine-insert at 16 amino acid positions (4 FWER p < 0.05; 12 q-value ≤ 0.20); significantly decreased with physicochemical-weighted Hamming distance to the vaccine-strain sequence for Spike, receptor-binding domain, N-terminal domain, and S1 (FWER p < 0.001); differed (FWER ≤ 0.05) by distance to the vaccine strain measured by 9 antibody-epitope escape scores and 4 NTD neutralization-impacting features; and decreased (p = 0.011) with neutralization resistance level to vaccinee sera. VE against severe–critical COVID-19 was stable across most sequence features but lower against the most distant viruses.

Initial SARS-CoV-2 vaccine candidates were based on the virus's original lineage, as represented by the index strain with Spike D614 (NC_045512; https://www.ncbi.nlm.nih.gov/nuccore/1798174254). As the virus has evolved, the efficacy of these vaccines against symptomatic infection has waned[1,2], and new vaccine inserts have been developed.

Based on data from a randomized, placebo-controlled vaccine efficacy (VE) trial on clinical outcomes and pathogen sequences isolated from participants experiencing clinical outcomes, sieve analysis assesses how VE depends on pathogen sequence features[3,4]. Pajon et al.[5] and Sadoff et al.[6] showed how the VE against symptomatic COVID-19 was lower against certain variants than against the Reference strain in the phase 3 COVE trial of two doses of Moderna's mRNA-1273 vaccine and the phase 3 ENSEMBLE trial of a single dose of Janssen's Ad26.COV2.S vaccine, respectively. [As in ref. 6, Reference is defined as the basal outbreak lineage B.1, which bears the D614G mutation.] Cao et al. showed that VE was higher in COVID-19 VE trials where circulating viruses had shorter Spike sequence

Hamming distances to the vaccine strain[7]. These sieve analyses only considered Spike viral variation defined by the WHO-defined variant category or the unweighted Spike protein distance. They did not assess how VE depends on other Spike sequence features, such as at the level of individual mutations or features that impact immunological functions such as anti-SARS-CoV-2 neutralization[8–13], relevant given the strong evidence of neutralizing antibodies (nAbs) as a cross-platform correlate of protection[14–16].

In this work, we report the results of a sieve analysis of the ENSEMBLE trial (NCT04505722), which enrolled over 40,000 participants and was conducted in Argentina, Brazil, Chile, Colombia, Mexico, Peru, South Africa, and the United States[6,17]. The sieve analysis considered baseline SARS-CoV-2 seronegative per-protocol participants and the primary endpoint (moderate to severe–critical COVID-19), as well as the severe–critical COVID-19 endpoint, during the double-blinded period of follow-up. The major conclusions of the current work are that in Latin America, where Spike diversity was greatest, VE differed by multiple Spike, receptor-binding domain

✉e-mail: pgilbert@fredhutch.org

(RBD), N-terminal domain (NTD), and S1 sequence features, as well as by distance to the vaccine strain as measured by multiple antibody-escape scores and neutralization-impacting features. Most of these significant sieve effects are linked to the Lambda lineage, implicating Lambda as a likely escape variant. Moreover, VE against severe–critical COVID-19 was generally stable across most sequence features, although it was lower against the most distant viruses.

## Results

### SARS-CoV-2 sequence data

A total of 1345 SARS-CoV-2 Spike amino acid sequences were obtained from 1224 participants experiencing the moderate to severe-critical primary endpoint. All sequences were variant-typed to either the Reference lineage or to one of nine different WHO-defined variants (Table 1, Fig. 1a, and Supplementary Table 5). In Latin America, lineages that circulated at the beginning of the study period, e.g., Reference, were closer to the sequence from the vaccine insert than later emerging lineages,

with Lambda the most distant (Fig. 1b and Supplementary Fig. 1). Similar results were obtained in South Africa and the United States (Supplementary Figs. 2, 3, respectively).

### Greatest SARS-CoV-2 Spike diversity in Latin America

Most sequences were obtained from participants in Latin America ($n = 776$), with additional sequences from the US ($n = 323$) and South Africa ($n = 125$) (Supplementary Table 6). Participant demographics of the sieve analysis cohort are shown in Table 2 (Latin America) and Supplementary Tables 7–9 (US, South Africa, all regions pooled, respectively). Five main variants circulated in Latin America (Reference, Zeta, Gamma, Lambda, Mu), while the South African sequences were 76% Beta and 17% Delta, and the US sequences were 85% Reference (Fig. 1a). There was greater Spike AA sequence diversity in Latin America compared to South Africa and the US (Rao's Q = 10.1 vs. 7.7 vs. 3.3, respectively; Supplementary Fig. 4).

The succession of distinct co-circulating variants in Latin America and the resulting broadest dynamic range of inter-individual sequence

**Table 1 | Numbers of primary endpoint COVID-19 cases with Spike amino acid sequence data by treatment arm, geographic region, and primary endpoint case lineage**

| Geographic Region | | | | | | | | |
|---|---|---|---|---|---|---|---|---|
| | Latin America | | South Africa | | United States | | Pooled | |
| Primary endpoint case lineage | Vaccine (329)[a] | Placebo (634) | Vaccine (62) | Placebo (110) | Vaccine (93) | Placebo (323) | Vaccine (484) | Placebo (1067) |
| Reference | 72 | 196 | 1 | 4 | 52 | 221 | 125 | 421 |
| Alpha | 4 | 10 | 1 | 2 | 4 | 16 | 9 | 28 |
| Beta | - | - | 36 | 59 | - | - | 36 | 59 |
| Delta | - | - | 11 | 10 | - | - | 11 | 10 |
| Epsilon | - | 2 | - | - | 8 | 15 | 8 | 17 |
| Gamma | 73 | 111 | - | - | 1 | - | 74 | 111 |
| Iota | - | - | - | - | - | 4 | 0 | 4 |
| Lambda | 43 | 45 | - | 1 | - | - | 43 | 46 |
| Mu | 38 | 57 | - | - | - | - | 38 | 57 |
| Zeta | 33 | 92 | - | - | 1 | 1 | 34 | 93 |
| No Sequence Obtained | 66 | 121 | 13 | 34 | 27 | 66 | 106 | 221 |

A primary endpoint case is defined as the moderate to severe-critical primary COVID-19 endpoint in the per-protocol baseline seronegative cohort, with disease onset starting 14 days post-vaccination through to a participant's unblinding date.

[a]Numbers in parentheses are numbers of moderate to severe-critical COVID-19 primary endpoints caused by the listed SARS-CoV-2 lineage, regardless of the availability of SARS-CoV-2 sequence data

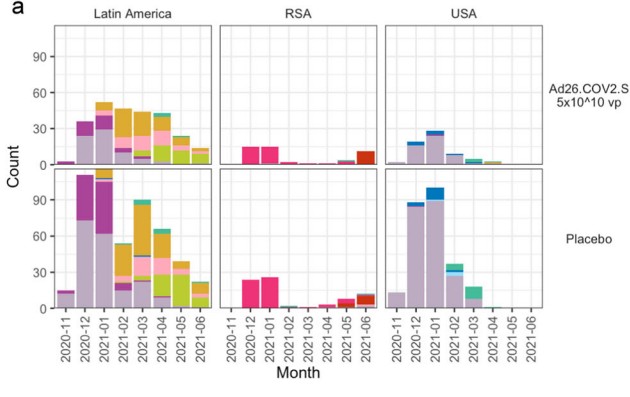

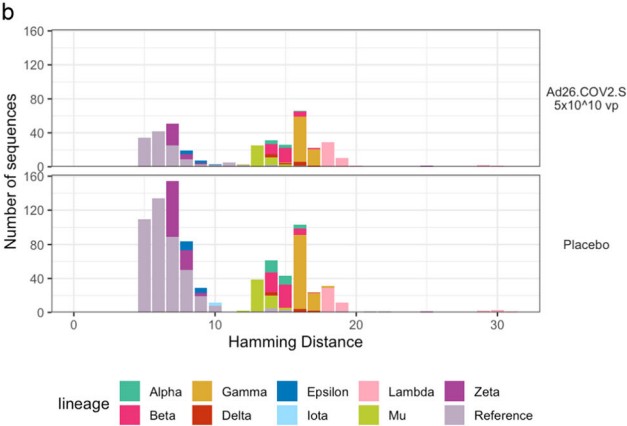

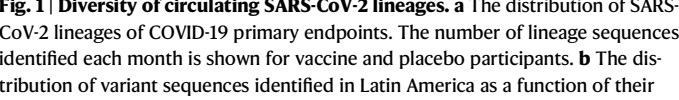

**Fig. 1 | Diversity of circulating SARS-CoV-2 lineages. a** The distribution of SARS-CoV-2 lineages of COVID-19 primary endpoints. The number of lineage sequences identified each month is shown for vaccine and placebo participants. **b** The distribution of variant sequences identified in Latin America as a function of their Spike Hamming distance from the vaccine insert. Lineages are color-coded as follows: Alpha, teal; Beta, pink; Delta, red; Epsilon, dark blue; Gamma, mustard; Iota, light blue; Lambda, pink; Mu, light green; Zeta, purple; and Reference, lavender. vp viral particles.

## Table 2 | Demographics of participants in Latin America in the sieve analysis cohort

| Characteristics | Vaccine (N = 329) | Placebo (N = 634) | Total (N = 963) |
|---|---|---|---|
| **Age** | | | |
| Age 18–59 | 261 (79.3%) | 523 (82.5%) | 784 (81.4%) |
| Age ≥60 | 68 (20.7%) | 111 (17.5%) | 179 (18.6%) |
| Mean (Range) | 43.4 (18, 83) | 44.3 (18, 83) | 44.0 (18, 83) |
| **Sex assigned at birth** | | | |
| Female | 139 (42.2%) | 270 (42.6%) | 409 (42.5%) |
| Male | 190 (57.8%) | 364 (57.4%) | 554 (57.5%) |
| **BMI** | | | |
| Underweight BMI <18.5 | 5 (1.5%) | 3 (0.5%) | 8 (0.8%) |
| Normal 18.5≤ BMI <25 | 103 (31.3%) | 194 (30.6%) | 297 (30.8%) |
| Overweight 25≤ BMI <30 | 72 (21.9%) | 154 (24.3%) | 226 (23.5%) |
| Obese BMI ≥30 | 149 (45.3%) | 282 (44.5%) | 431 (44.8%) |
| - | - | 1 (0.2%) | 1 (0.1%) |
| **Ethnicity** | | | |
| Hispanic or Latino | 319 (97.0%) | 611 (96.4%) | 930 (96.6%) |
| Not Hispanic or Latino | 6 (1.8%) | 19 (3.0%) | 25 (2.6%) |
| Unknown | 4 (1.2%) | 4 (0.6%) | 8 (0.8%) |
| **Race** | | | |
| American Indian Or Alaska Native | 101 (30.7%) | 187 (29.5%) | 288 (29.9%) |
| Asian | 5 (1.5%) | 2 (0.3%) | 7 (0.7%) |
| Black Or African American | 12 (3.6%) | 22 (3.5%) | 34 (3.5%) |
| White | 146 (44.4%) | 299 (47.2%) | 445 (46.2%) |
| Multiple | 50 (15.2%) | 104 (16.4%) | 154 (16.0%) |
| Other | 15 (4.6%) | 20 (3.2%) | 35 (3.6%) |
| **Country** | | | |
| Argentina | 45 (13.7%) | 102 (16.1%) | 147 (15.3%) |
| Brazil | 109 (33.1%) | 212 (33.4%) | 321 (33.3%) |
| Chile | 6 (1.8%) | 9 (1.4%) | 15 (1.6%) |
| Colombia | 105 (31.9%) | 209 (33.0%) | 314 (32.6%) |
| Mexico | 3 (0.9%) | 9 (1.4%) | 12 (1.2%) |
| Peru | 61 (18.5%) | 93 (14.7%) | 154 (16.0%) |
| **Risk for severe COVID-19[a]** | | | |
| At-risk | 114 (34.7%) | 211 (33.3%) | 325 (33.7%) |
| Not at-risk | 215 (65.3%) | 423 (66.7%) | 638 (66.3%) |
| **Age, risk for severe COVID-19** | | | |
| Age 18–59 at-risk | 75 (22.8%) | 152 (24.0%) | 227 (23.6%) |
| Age 18–59 not at-risk | 186 (56.5%) | 371 (58.5%) | 557 (57.8%) |
| Age ≥60 at-risk | 39 (11.9%) | 59 (9.3%) | 98 (10.2%) |
| Age ≥60 not at-risk | 29 (8.8%) | 52 (8.2%) | 81 (8.4%) |
| **HIV status** | | | |
| Negative | 324 (98.5%) | 623 (98.3%) | 947 (98.3%) |
| Living with HIV | 5 (1.5%) | 11 (1.7%) | 16 (1.7%) |

[a] "At-risk" is defined as having one or more comorbidities [listed in ref. 17] associated with elevated risk of severe COVID-19.

diversity, and the greatest number of COVID-19 endpoints, implies that sieve analyses of the Latin America region have the greatest statistical power. In contrast, the domination of the Reference lineage in the United States and the Beta and Delta lineages in South Africa constrained the sequence diversity's dynamic range and limited the power of these sieve analyses. Therefore, we focus on the results from Latin America, with the United States and South Africa results reported in Supplementary Note 1.

### Differential vaccine efficacy by SARS-CoV-2 lineage

All reported results on VE by SARS-CoV-2 features are based on feature-specific proportional-hazards models[18,19] [see the Statistical Analysis Plan (SAP, provided in ref. 20 and at the end of the Supplementary Information)]. Figure 2a shows VE against the primary COVID-19 endpoint caused by the Reference, Gamma, Zeta, Lambda, and Mu lineages, and Fig. 2b shows VE against the primary COVID-19 endpoint caused by the groupings of all other lineages excluding each individual lineage (referred to as not-lineage). Figure 2c shows differential VE against pairs of lineages or against pairs of lineage vs. not-lineage. VE was significantly higher against Reference than against Lambda and against not-reference lineages [family-wise error rate (FWER) $p < 0.05$]. It was also significantly higher against not-Lambda vs. Lambda and against Zeta vs. Lambda (FWER $p ≤ 0.05$), and higher against Reference vs. Gamma, Reference vs. Mu, Zeta vs. Gamma, and Zeta vs. Mu ($q$ value ≤0.20). Country-specific results for the same analyses are shown in Supplementary Figs. 5–7, where due to reduced sample size, the confidence intervals are generally much wider, and fewer pairwise comparisons of VE against lineages could be performed.

### Higher vaccine efficacy against vaccine-matched AA residues

We scanned across all Spike AA positions with sufficient residue variability (at least 20 endpoints with a vaccine-mismatched residue: $n = 37$ positions). VE significantly differed ($q$ value ≤0.20) by residue match vs. mismatch to the vaccine strain residue at 16 positions (Fig. 2d; four positions with FWER $p ≤ 0.05$: 75, 76, 253, 490). Similarly, when assessing the presence or absence of specific residues at each AA position, VE significantly differed ($q$ value ≤0.20) for 38 residues/residue features (75V vs. not-75V and 76I vs. not-76I with FWER $p ≤ 0.05$) distributed across these 16 positions (Fig. 2e). Thirteen of these 16 positions harbored characteristic mutations of the Lambda variant and not for any other variants, and very highly covaried with Lambda vs. not-Lambda (Supplementary Fig. 8, Mstar[21] >0.85), thereby providing nearly equivalent signatures of differential VE captured by Lambda vs. not-Lambda. The full results of the covariability analysis are in Supplementary Note 1. Four of the 1277 analyzed AA positions (417, 452, 484, 490) were prespecified as being hypothesized to impact neutralization based on an association with a reduced nAb response in mRNA vaccine recipients[22–24], or evidence for increased transmissibility (452)[22] or increased infectivity in vitro (452, 490)[22,24,25]. Of these, positions 452 and 490 were found to significantly impact VE (FWER $p ≤ 0.05$). Latin America country-specific results for the same analyses are shown in Supplementary Figs. 9–12, where due to reduced sample size fewer AA positions were screened in for assessment of differential VE.

Supplementary Figs. 13–15 provide complete results, including by geographic region.

### Vaccine efficacy decreases with vaccine-insert distance

VE significantly decreased with physicochemical-weighted Hamming distance (between the observed vs. vaccine insert sequence) for Spike, RBD, NTD, and S1 (Fig. 3, FWER $p < 0.001$) but not for S2 ($p = 0.78$). Against viruses with shortest Spike distances (average six residue mismatches), VE was 69% (95% CI: 60 to 76%), and against viruses with 25th, 50th, 75th, and 95th percentile Spike distances (average 8.1, 12.9, 17.8, 18.6 residue mismatches), VE was 64% (56, 71%), 52% (44, 58%), 34% (19, 46%), and 30% (13, 44%), respectively. The median distances of sequences for vaccine:placebo were 15.0:9.5 for Spike, 2.6:1.0 for RBD, 4.0:1.6 for NTD, 11.7:6.2 for S1, and 3.1:3.2 for S2. A sensitivity analysis of VE by physicochemical-weighted Hamming distance performed after removing the Lambda and Zeta outliers (defined by distance >20) in Fig. 3 yielded similar results (Supplementary Fig. 16). Supplementary Tables 10 and 11 show inferences about differences in

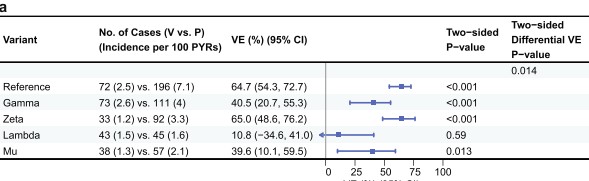

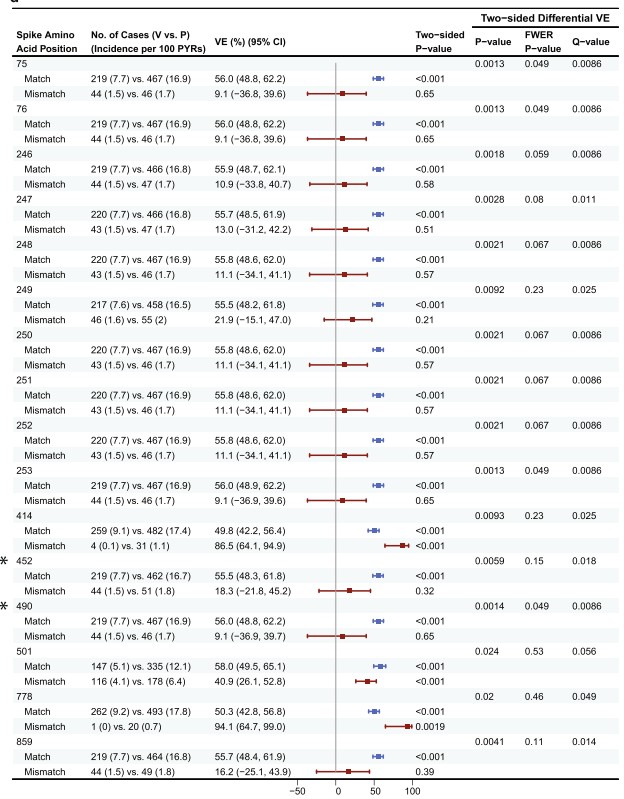

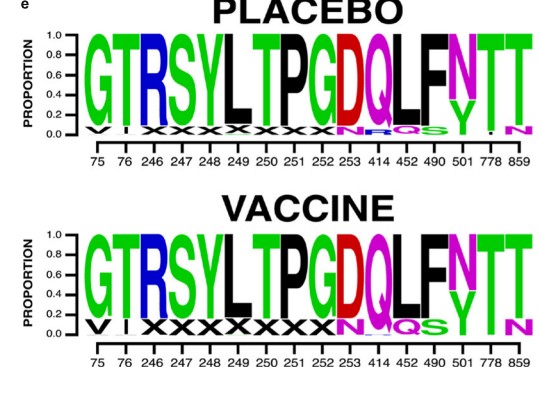

**Fig. 2 | For the Latin America cohort, differential vaccine efficacy (VE) by lineage or by residue feature. a** VE estimates against the primary COVID-19 endpoint caused by SARS-CoV-2 lineages (lineage "X"). **b** VE estimates against the primary COVID-19 endpoint caused by all other lineages combined ("Not X"). **c** Differential VE estimates against the primary COVID-19 endpoint across pairs of lineages or across a lineage ("X") vs. all other lineages ("Not X"). **d** VE estimates against the primary COVID-19 endpoint caused by SARS-CoV-2 with a vaccine-matched or vaccine-mismatched residue at each of the 16 Spike amino acid (AA) residues with differential VE ($q$ value <0.2 and unadjusted $p \leq 0.05$). Results for VE against matched residue genotypes are shown in blue and for mismatched residue genotypes in maroon. The two amino acid positions hypothesized to impact VE

(452 and 490)[22,24,25] are identified with an asterisk. In the forest plots in **a**, **b**, and **d**, the solid squares represent the VE point estimates, and the error bars display the 95% confidence intervals. In panels **a**, **b**, and **d**, the "Two-sided $P$ value" is from testing the null hypothesis $H_{Aj0}$ vs. $H_{Aj2}$ using the test statistic $U_{2j}$ (pp 17–18 of Heng et al.[18].). In panels **a**, **c**, and **d**, the "two-sided differential VE $P$ value" is from testing the null hypothesis $H_{B0}$ vs. $H_{B2}$ using the test statistic $T_2$ (pp 18–19 of Heng et al.[18].). **e** Logo plots showing the distributions of AA residues and gaps/deletions (represented by "X") by treatment arm at the 16 positions in (**d**). AA residues are color-coded by their chemical properties: green, polar; red, acidic; blue, basic; black, hydrophobic. CI confidence interval, DVE differential vaccine efficacy, FDR false discovery rate, FWER family-wise error rate, PYRs person-years.

mean distances of vaccine vs. placebo sequences. Supplementary Figs. 17–21 and Supplementary Table 12 provide complete results, including by geographic region, where Supplementary Table 12 shows that VE decreased with weighted Hamming distance for RBD, NTD, and S1 in the US ($q$ value ≤0.20).

By lineage, ordered by placebo arm COVID-19 endpoint Spike distance to the vaccine strain, Reference viruses had 6.0–17.7 residue mismatches, Zeta 8.1–22.1 mismatches, Epsilon 10.7 mismatches, Mu 12.2–16.8 mismatches, Alpha 14.5–16.8 mismatches, Gamma 16.7–20.2 mismatches, and Lambda 17.2–27.7 mismatches. This ordering of lineages by protein distance matches the ordering of the VE estimates by lineage category, suggesting that overall Spike evolution is a reasonable metric capturing VE decline with a variant. The results are generally similarly ordered for the RBD, NTD, and S1 distances (Supplementary Fig. 22).

## Vaccine efficacy decreases with antibody-escape score

Neutralization-relevant RBD features were defined where mutations impact binding in deep mutational scanning (DMS) experiments[26] (see Supplementary Methods). Escape scores were defined for all antibodies and for each of 10 epitope-specific clusters of AA sites (see Methods), labeled DMS (all antibodies) and DMS1 through DMS10. Vaccine efficacy significantly decreased ($q$ value ≤0.20) with each of the DMS, DMS2, DMS6, DMS7, and DMS8 escape scores (FWER $p \leq 0.05$) as well as for DMS1, DMS5, DMS9 ($q$ value ≤0.20 and FWER >0.05) (Supplementary Table 13). Supplementary Tables 14 and 15 show the mean DMS escape scores in the vaccine arm and in the placebo arm of the disease-causing SARS-CoV-2 isolates, as well as the difference in mean DMS escape score between the two arms. To accommodate for missing sequences, in the analysis whose results are shown in Supplementary Table 14, doubly robust targeted minimum loss-based estimation was used; in

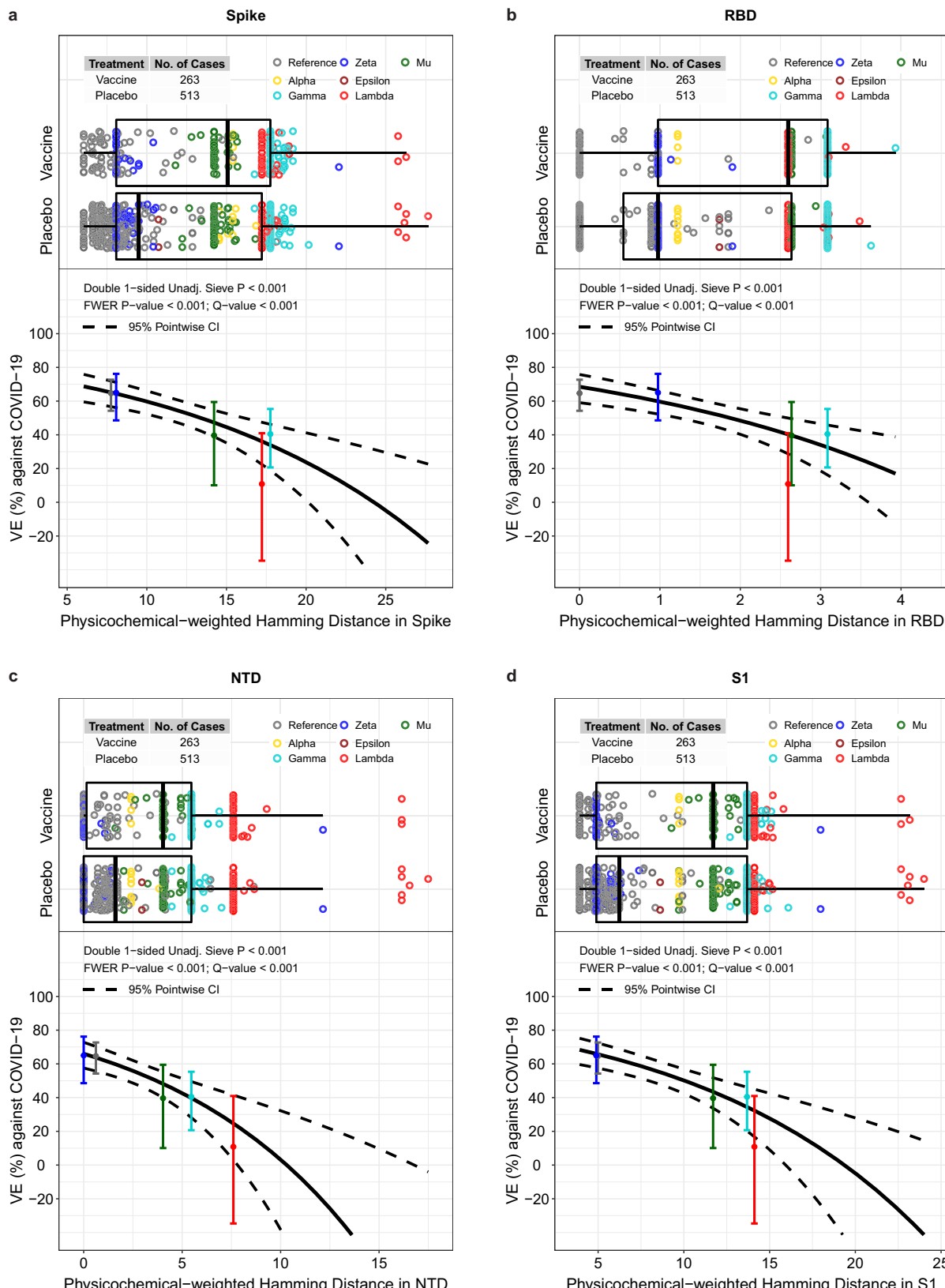

Supplementary Table 15, inverse probability weighting was used. Results were generally similar using the two different statistical methods, with greater mean DMS escape scores in the vaccine arm than the placebo arm for all of the clusters and the lower limit of the 95% CI usually greater than zero. The greatest difference in mean DMS escape score (vaccine – placebo) was seen for DMS5 [0.051 (0.0032, 0.098) in Supplementary

Table 14, 0.13 (0.073, 0.19) in Supplementary Table 15. Geographic region-specific results are also shown in Supplementary Tables 14 and 15.

Alternatively, we defined putative antibody footprint site sets (including whole Spike) based on structures of SARS-CoV-2 in complex with antibodies available from the Protein Data Bank (PDB). Each sequence was assigned an escape score based on a class of epitopes

**Fig. 3 | For the Latin America cohort, vaccine efficacy (VE) against the primary COVID-19 endpoint by different physicochemical-weighted (PCW) Hamming distances.** The Hamming distances were from the disease-causing SARS-CoV-2 isolate to that of the vaccine-insert sequence, for the following regions: **a** Spike, **b** the RBD domain, **c** the NTD domain, or **d** the S1 region. The top plot in each panel shows the distributions of distances by treatment arm, color-coded by lineage: Alpha, yellow; Epsilon, brown; Gamma, turquoise; Lambda, red; Mu, green; Reference, gray; Zeta, blue. The left and right edges of the box plots represent the 25th and 75th percentiles of PCW-Hamming distance in the designated region, and the vertical middle line represents the 50th percentile. The horizontal bars extend from the 25th (or 75th) percentile of PCW-Hamming distance to the minimum (or maximum) PCW-Hamming distance within the 25th (or 75th) percentile of Hamming distance minus (or plus) 1.5 times the interquartile range. The bottom plot in each panel shows the estimated VE by SARS-CoV-2 sequence distance. The dotted lines

are pointwise 95% confidence intervals. The dots are overall VE estimates for the given lineage placed at the lineage-specific median distance of placebo arm endpoints, with vertical bars indicating their pointwise 95% confidence intervals. In each panel, the "Double 1-sided unadjusted sieve $p$ value" doubles the $p$ value from a one-sided Wald test of the null hypothesis of constant VE vs. the alternative hypothesis of a decreasing VE with an increasing value of the feature on the x-axis (Juraska and Gilbert[19], Section 5). Two Zeta sequences are visible outliers from other Zeta sequences; both sequences have two large deletions (9AA and 7AA in length) in the N-terminal domain. The plots reveal that Lambda has two sub-lineages, one ($n = 79$) with a range of distances 17.2–18.9 and a second ($n = 9$) with a range of distances 25.8–27.7, due to a 13-AA deletion between sites 64 and 76. CI confidence interval, FWER family-wise error rate, NTD N-terminal domain, RBD receptor-binding domain, Unadj. unadjusted.

(see Supplementary Methods). These features are referred to as PDB1 through PDB14, with the first 12 clusters in the RBD and PDB13 and PDB14 in the NTD. Vaccine efficacy significantly decreased ($q$ value ≤0.20) with the escape scores for PDB4, PDB7, PDB8, and PDB13 (FWER $p \le 0.05$) as well as for PDB1 and PDB3 ($q$ value ≤0.20 and FWER >0.05) (Supplementary Table 16). Supplementary Tables 17 and 18 show the mean PDB escape scores in the vaccine arm and in the placebo arm of the disease-causing SARS-CoV-2 isolates, as well as the difference in mean PDB escape score between the two arms. Analyses were performed the same as for Supplementary Tables 14 and 15. For each cluster, the mean PDB escape score was generally higher in the vaccine arm than in the placebo arm. The greatest difference in mean PDB escape score (vaccine − placebo) was seen for PDB13 [0.27 (0.043, 0.5) in Supplementary Table 17 and 0.47 (0.24, 0.7) in Supplementary Table 18]. Geographic region-specific results are also shown in Supplementary Tables 17 and 18.

To interpret the DMS and PDB results, we focus on the epitope-specific features with FWER $p \le 0.05$ that carry the greatest amount of independent information based on inter-correlation and hierarchical clustering analysis (Supplementary Note 1 and Supplementary Figs. 23, 24): DMS2, PDB7, PDB8, and PDB13. The sieve analysis results are similar across these four features, with estimated VE at 60–70% against viruses with an escape score of zero and decreasing to 0%–20% against viruses with a maximum escape score. PDB8 and PDB13 rank highest for discriminating VE with a slightly greater span of VE point estimates over the range of escape scores (spans 20–60%, 16–60%, 21–69%, and 1–57% for DMS2, PDB7, PDB8, and PDB13, respectively) (Fig. 4a–d).

Figure 5a lists the Spike AA residues in each epitope footprint, and the visualizations in Fig. 5b–e show the positions comprising the four antibody-epitope footprints on a Spike monomer structure. Supplementary Figs. 25–33, 34–40 provide complete results for DMS and PDB features, respectively. Another reason PDB8 was highlighted is its balanced contacts across the whole receptor-binding motif (RBM) whereas the other RBM-specific clusters (PDB1–PDB6) are more tightly grouped within a region of the RBM (Fig. 5f). The epitope of PDB8 is better to monitor broadly the mutations in the RBM than PDB1–6, which have more limited targets. Among the non-RBM focusing antibodies (PDB7, PDB9–PDB14), PDB7 (which includes variable sites) and PDB13 correspond to the most accessible sites on Spike in a closed prefusion trimer (Fig. 5g). These sites are relatively variable among SARS-CoV-2 sequences. The PDB9, PBD10, and PDB11 epitopes are cryptic and conserved, whereas the epitope for PDB12 appears accessible and conserved. Among NTD antibodies, the Fc domain may be more accessible for PDB13 than for PDB14 (Fig. 5h), which may be linked to improved Fc-dependent virus clearance.

**Vaccine efficacy decreases with NTD-linked reduced neutralization**

Seven dichotomous NTD features (Methods) were assessed for a sieve effect as for vaccine-match vs. vaccine-mismatch binary features. Six of

the 7 NTD features significantly impacted VE ($q$ value ≤0.20): NTD4, NTD6, NTD1, NTD3, NTD5, and NTD7 (where the last four also had FWER $p \le 0.05$) (Fig. 6a). Figure 6b–e show the spatial locations in the NTD of the four features that impacted VE (FWER $p \le 0.05$).

**Vaccine efficacy decreases by neutralization resistance score**

All of the sieve analyses study how VE depends on Spike AA features except one: a neutralization sieve analysis that scores each virus's lineage by its experimentally measured sensitivity to neutralization by Ad26.COV2.S vaccinee sera[27,28]. VE decreased with this variant-neutralization resistance score ($p = 0.011$) (Fig. 7). Under one model for the neutralization assay being a perfect correlate of protection, the estimates of VE for each of the five lineages would fall on the curve of VE by variant-neutralization resistance score. Lambda had evidence of deviating from the curve, with VE 55% (48, 62%) when estimated based on measured neutralization sensitivity compared to VE 11% (−35, 41%) when estimated based on direct analysis of Lambda ignoring neutralization data. In contrast, the weighted Hamming distance analyses yielded VE estimates at Lambda-variant distance values that are closer to the VE 11% figure.

Supplementary Fig. S41 provides complete results by geographic region.

**Multivariable virus features as predictors of treatment arm**

A variable importance measure (VIM) analysis by ensemble machine learning[29] was conducted to assess how well different groups of virus features in COVID-19 endpoint cases predicted treatment arm beyond that provided by baseline risk factors (whether the COVID-19 endpoint was from Colombia and enrollment periods). Virus features defined based on neutralization data were the top-performing predictors of the treatment arm, with the DMS RBD antibody-escape features having the highest estimated VIM (0.073, $p$ value for a test of the null hypothesis of zero VIM = 0.043; Supplementary Fig. 42).

The second-most important classifying variables were the set of all nAb correlate of protection (CoP) hypothesis features, defined as DMS RBD and PDB antibody-escape features, NTD neutralization-relevant features, and the variant-neutralization sensitivity score (VIM = 0.051); PDB antibody-escape features (VIM = 0.049); and the variant-neutralization sensitivity score (VIM = 0.048) (Supplementary Fig. 42). The unbiased features specified to include all Spike AA variation ignoring neutralization hypotheses had the lowest estimated variable importance (VIM = 0.036 to 0.046 for the weighted Hamming distances and Spike AAs).

**Generally stable vaccine efficacy against severe-critical COVID-19**

Differential VE against severe-critical COVID-19 by lineage could only be assessed for Latin America, with VE of 83% (64, 92%) against Reference, 64% (26, 83%) against Gamma, 94% (−27, 100%) against Zeta, 62% (−31%, 89%) against Lambda, and 84% (42, 96%) against Mu

**a**

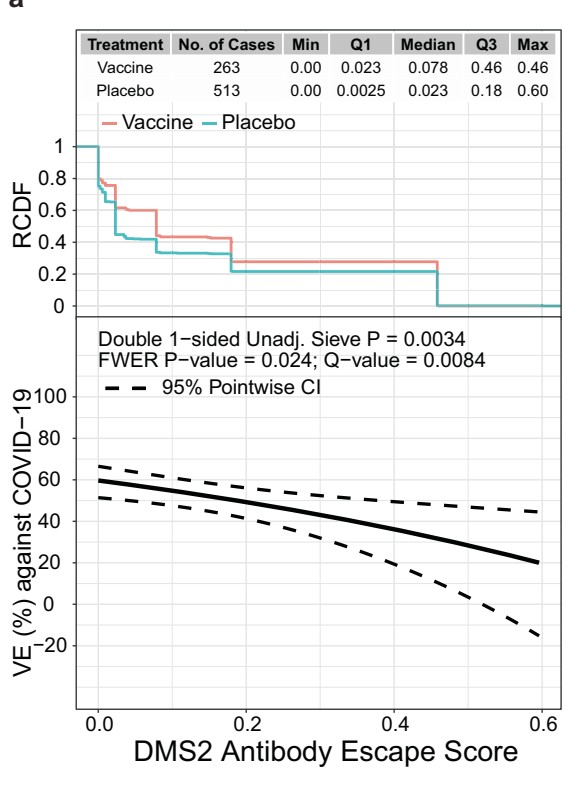

**b**

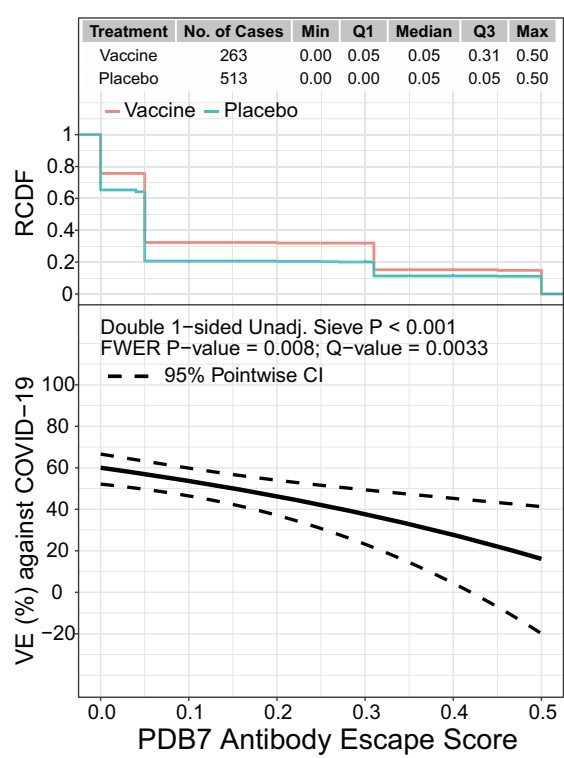

**c**

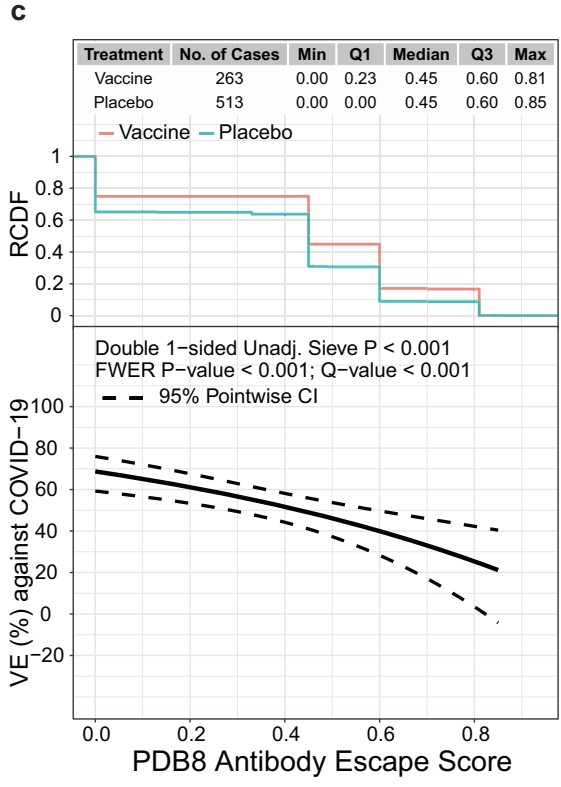

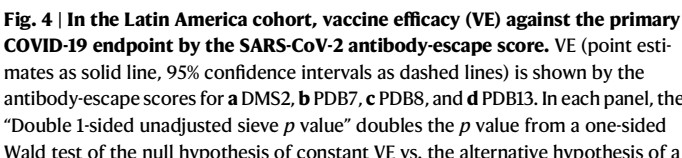

**d**

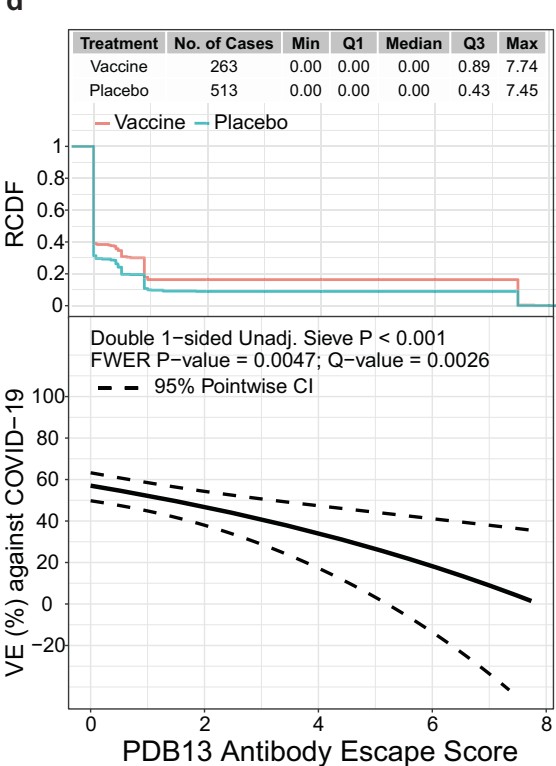

**Fig. 4 | In the Latin America cohort, vaccine efficacy (VE) against the primary COVID-19 endpoint by the SARS-CoV-2 antibody-escape score.** VE (point estimates as solid line, 95% confidence intervals as dashed lines) is shown by the antibody-escape scores for **a** DMS2, **b** PDB7, **c** PDB8, and **d** PDB13. In each panel, the "Double 1-sided unadjusted sieve *p* value" doubles the *p* value from a one-sided Wald test of the null hypothesis of constant VE vs. the alternative hypothesis of a decreasing VE with an increasing value of the feature on the x-axis (Juraska and Gilbert[19], Section 5). The plot at the top of each panel shows the reverse cumulative distribution function (RCDF) of the relevant antibody-binding escape score across SARS-CoV-2 isolates by treatment arm: Vaccine, pink; Placebo, turquoise. CI confidence interval, DMS deep mutational scanning, FWER family-wise error rate, PDB Protein Data Bank, Unadj. unadjusted.

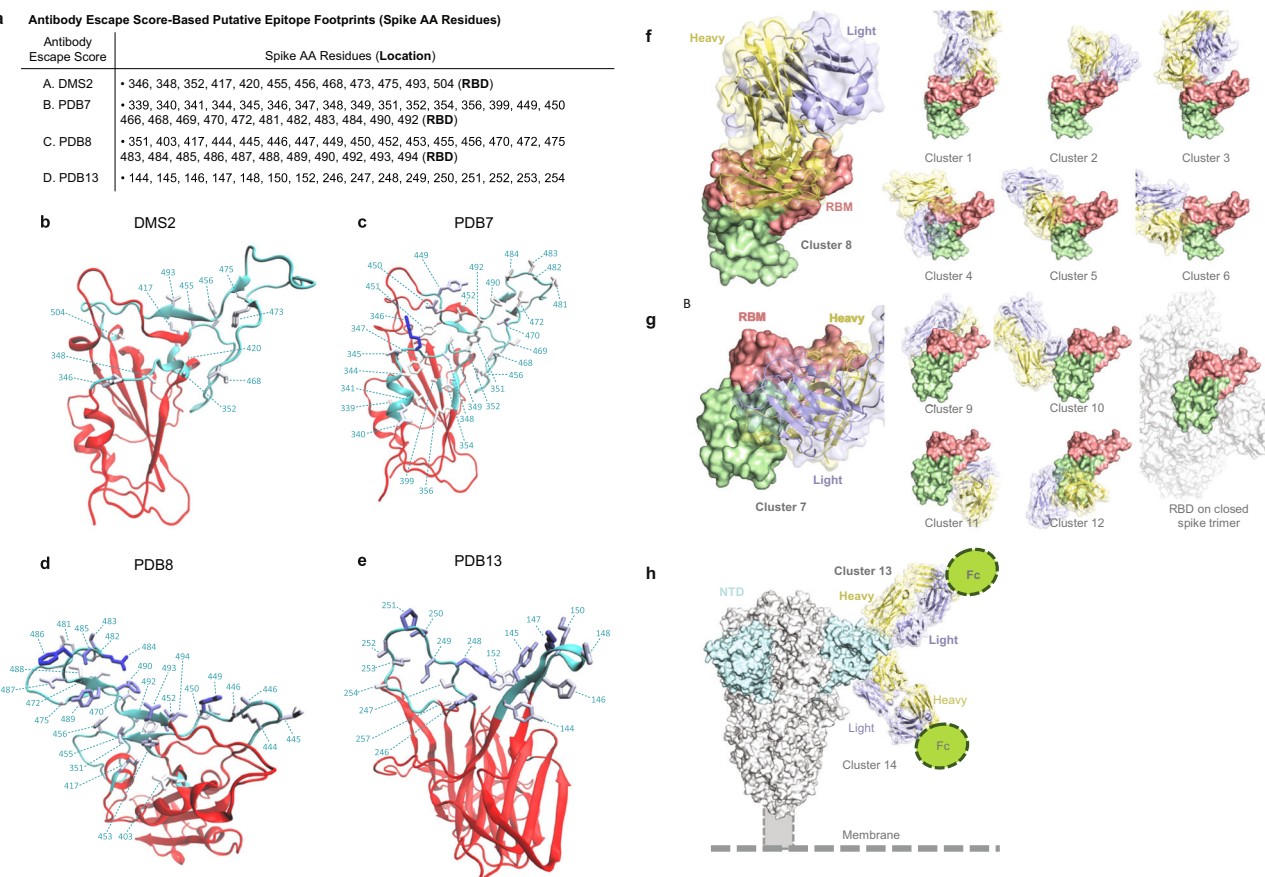

**Fig. 5 | Structural visualizations of antibody footprints in Spike on antibody-escape score clusters and neutralization features where a sieve effect was seen.** **a** Spike amino acid (AA) residues constituting each antibody-escape score-based putative epitope footprint. **b–e** For each set of residues constituting an antibody-epitope footprint for DMS2, PDB7, PDB8, and PDB13, the image shows the set of AA positions comprising the footprint on a Spike monomer NTD or RBD structure. Cyan ribbons highlight epitope footprint residues, while red ribbons make up the rest of RBD [**b** DMS2, **c** PDB7, and **d** PDB8)] or NTD (**e** PDB13). Residue numbers and cyan dashed lines are used to label footprint residues. Each structure's orientation was chosen to best visualize all residues of a footprint. Residues are colored based on their cluster weights going from white to blue with increasing weight. **f–h** Epitope location of representative clusters 8, 7, and 13. Color coding denotes the following: Receptor-binding motif, red; heavy chain, yellow; light chain, lavender; N-terminal domain, turquoise; clusters, green. DMS deep mutational scanning, NTD N-terminal domain, PDB Protein Data Bank, RBD receptor-binding domain.

(Supplementary Table 19). There was no evidence of variation in VE across the lineages ($p = 0.50$) (Supplementary Tables 19, 20). The estimates of VE were similar/stable across AA positions with vaccine-matched vs. vaccine-mismatched residue, with all unadjusted $p$ values for differential VE above 0.05 (Supplementary Fig. 43). For the key positions 452 and 490 found to show sieve effects for the primary COVID-19 endpoint, the results for the severe-critical COVID-19 endpoint were VE 79% (68, 87%) against 452-matched virus compared to VE 70% (3, 91%) against 452-mismatched virus ($p = 0.58$ for difference), and VE 80% (68, 87%) against 490-matched virus compared to VE 62% (−31, 89%) against 490-mismatched virus ($p = 0.34$ for differential VE). For the DMS antibody-escape score distances, the data support stable VE across the distances (Supplementary Table 21). Similarly, the data support stable VE across RBD and PDB Spike-antibody-escape scores (Supplementary Table 22). VE was stable by variant-neutralization resistance score, with VE = 84% (67%, 92%) for the most sensitive lineage (ancestral) and VE = 73% (50, 85%) for the least sensitive lineage (Mu) ($p = 0.33$, Supplementary Fig. 44).

There was a trend of VE against severe–critical COVID-19 decreasing with the weighted Hamming distance for the Spike, NTD, and S1 regions ($q$ values = 0.20) (Supplementary Table 23 and Supplementary Figs. 45, 47, 48). The point estimates of VE suggested moderate declines of VE with distances. For example, the VE for Spike was 87% (71%, 94%) against viruses with a shortest distance of 6 and

66% (34%, 83%) against viruses with a long distance of 20 ($p = 0.12$). Supplementary Figs. 45–49 and Supplementary Table 24 provide complete information by geographic region. In addition, while VE was stable across levels of NTD1 through NTD4 ($p > 0.20$), it differed by levels of NTD5, NTD6, and NTD7, with VE of 61% (31, 78%) vs. 88% (76, 94%) for the two NTD5 genotypes ($q = 0.10$ for difference), VE of 60% (20, 80%) vs. 84% (72, 91%) for the two NTD6 genotypes ($q = 0.12$ for difference), and VE of 64% (32, 80%) vs. 85% (73, 92%) for the two NTD7 genotypes ($q = 0.12$ for difference) (Supplementary Table 24).

**Exploratory analyses of Lambda's escape and driver mutations**
A reviewer suggested additional analyses in pursuit of understanding how the level of VE against Lambda was lower than expected based on its level of neutralization resistance to vaccinee sera (Fig. 7) and the potential impact of driver vs. bystander mutations. Based on the reviewer's suggestions, we conducted two post-hoc exploratory analyses of Latin American data. First, we repeated the AA position site-scanning analysis, except removing all Lambda lineage viruses. Of the 16 AA positions with sieve effect evidence (Fig. 2d), three qualified for the sensitivity analysis based on sufficient residue variability (sites 414, 501, 778). The results of this sensitivity analysis also supported a sieve effect for all three positions (Supplementary Fig. 50), with VE estimates against vaccine-matched COVID-19 and against vaccine-mismatched COVID-19 in the Lambda-excluded analysis very similar to the all-

**a**

| NTD Features | No. of Cases (V vs. P) (Incidence per 100 PYRs) | VE (%) (95% CI) | | Two−sided P−value | Two−sided Differential VE | | |
|---|---|---|---|---|---|---|---|
| | | | | | P−value | FWER P−value | Q−value |
| NTD1 | | | | | 0.0016 | 0.0065 | 0.0025 |
| Mark Value = 1 | 93 (3.3) vs. 126 (4.6) | 31.0 (10.3, 46.8) | | 0.0055 | | | |
| Mark Value = 0 | 170 (6) vs. 387 (14) | 58.7 (51.0, 65.3) | | <0.001 | | | |
| NTD3 | | | | | 0.0017 | 0.0065 | 0.0025 |
| Mark Value = 1 | 46 (1.6) vs. 50 (1.8) | 12.5 (−30.1, 41.2) | | 0.51 | | | |
| Mark Value = 0 | 217 (7.6) vs. 463 (16.7) | 56.1 (48.9, 62.2) | | <0.001 | | | |
| NTD4 | | | | | 0.097 | 0.19 | 0.12 |
| Mark Value = 1 | 133 (4.7) vs. 223 (8.1) | 45.3 (33.0, 55.4) | | <0.001 | | | |
| Mark Value = 0 | 130 (4.6) vs. 290 (10.5) | 57.2 (47.9, 64.9) | | <0.001 | | | |
| NTD5 | | | | | <0.001 | 0.0021 | 0.0012 |
| Mark Value = 1 | 132 (4.6) vs. 188 (6.8) | 35.2 (19.8, 47.6) | | <0.001 | | | |
| Mark Value = 0 | 131 (4.6) vs. 325 (11.7) | 61.8 (53.7, 68.6) | | <0.001 | | | |
| NTD6 | | | | | 0.12 | 0.19 | 0.12 |
| Mark Value = 1 | 77 (2.7) vs. 120 (4.3) | 41.5 (22.7, 55.7) | | <0.001 | | | |
| Mark Value = 0 | 186 (6.5) vs. 393 (14.2) | 55.2 (47.1, 62.0) | | <0.001 | | | |
| NTD7 | | | | | <0.001 | 0.0013 | 0.0012 |
| Mark Value = 1 | 117 (4.1) vs. 157 (5.7) | 31.7 (14.2, 45.7) | | 0.0011 | | | |
| Mark Value = 0 | 146 (5.1) vs. 356 (12.9) | 61.0 (53.2, 67.5) | | <0.001 | | | |

−50 0 50 100
VE (%) (95% CI)

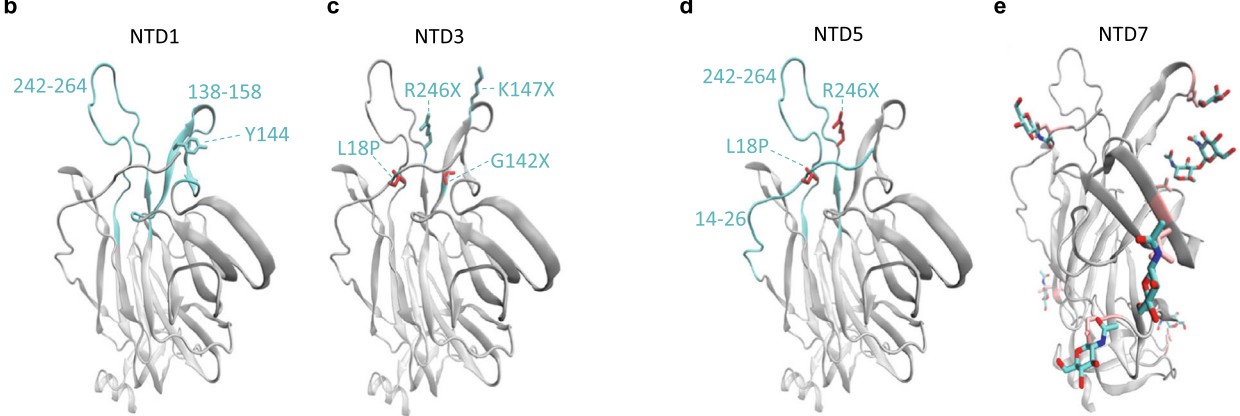

**Fig. 6 | In the Latin America cohort, N-terminal domain (NTD) sequence feature sieve analysis and visualization of defining sites in the NTD features that impacted vaccine efficacy (VE). a** VE estimates against the primary COVID-19 endpoint caused by SARS-CoV-2 with (vs. without) a NTD feature value, screened in as a specific hypothesis-driven neutralizing antibody (nAb) correlate of protection. VE estimates against SARS-CoV-2 harboring the NTD feature value are shown in blue; those against SARS-CoV-2 without the NTD feature value are shown in maroon. The solid squares represent the VE point estimates, and the error bars display the 95% confidence intervals. The "two-sided *P* value" is from testing the null hypothesis $H_{Aj0}$ vs. $H_{Aj2}$ using the test statistic $U_{2j}$ (pp 17–18 of Heng et al.[18].) and the "two-sided differential VE *P* value" is from testing the null hypothesis $H_{B0}$ vs. $H_{B2}$ using the test statistic $T_2$ (pp 18–19 of Heng et al.[18].). **b**–**e** Visualization of the constituent mutation sites that define the NTD feature, for each NTD features that significantly impacted VE with an FWER *p* < 0.05 (NTD1, NTD3, NTD5, and NTD7) (PDB 7L2C). **b** NTD1: Cyan colors the two insertions or deletions regions with one common residue for deletion (Y144) shown. **c** NTD3: The four specific mutations are labeled and colored cyan or red based on being mutations of interest or not respectively. **d** NTD5: Cyan colors the two substitution regions highlighting the two exclusions that are both found in this structure. **e** NTD7: Pink colors all asparagine residues that form N-linked glycans with the corresponding carbohydrate colored by atom (C: cyan, N: blue, O: red). Each structure shows the entire NTD region and colors all non-highlighted portions gray. CI confidence interval, FWER family-wise error rate, PYRs person-years.

lineage results shown in Fig. 2d. Therefore, the association of AAs at these positions with VE did not depend on the lineage being Lambda.

Secondly, as not accounting for ancestral relationships among sequences can potentially detect associations of AA changes with VE that are due to bystander mutations within distinct lineages, we conducted phylogenetically corrected GenSig analysis[30–34] of the 37 screened-in sites to better understand vaccine associations with residues (Supplementary Tables 25–27). This method, based on phylogenetic reconstruction from the nucleotide sequences (as both silent and non-silent mutations are accounted for in the imputed ancestral evolution of the lineages), scans all amino acid positions throughout the full Spike protein to assess whether ancestral changes to or away from each residue are associated with vaccination status. Of the 16 AA positions significantly associated with VE in Latin America (Fig. 2d), the Lambda variant mutation RSYLTPGD246-253N, RSYLTPGD with the N246 glycan in the NTD supersite epitope, was

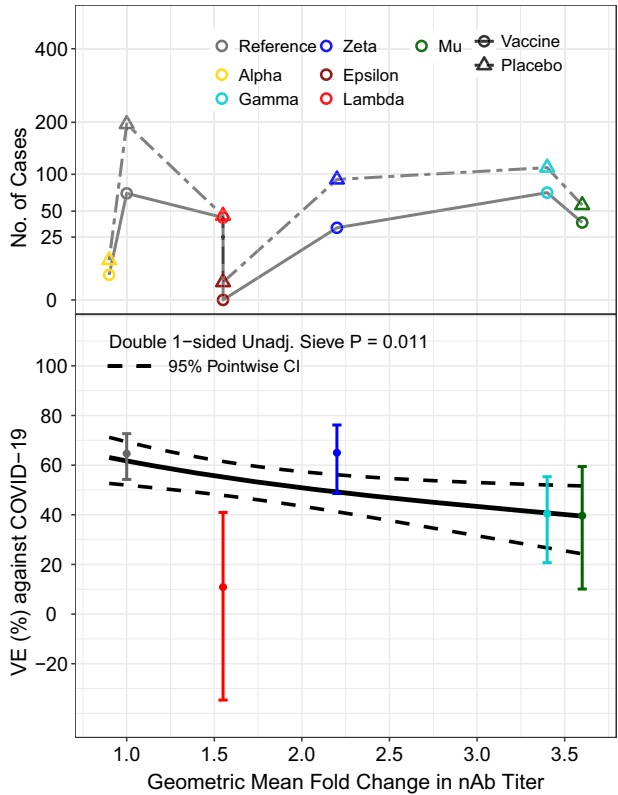

**Fig. 7 | In the Latin America cohort, neutralization phenotype sieve analysis.**
Vaccine efficacy (VE) against the primary COVID-19 endpoint is shown by geometric fold change in neutralizing antibody titer against the disease-causing SARS-CoV-2 variant vs. against the D614G Reference strain. The top plot shows the numbers of cases by treatment arm (Vaccine: open circle; Placebo: open triangle) and color-coded by lineage: Alpha, yellow; Epsilon, brown; Gamma, turquoise; Lambda, red; Mu, green; Reference, gray; Zeta, blue. The bottom plot shows the estimated vaccine efficacy by geometric fold change in nAb titer against the disease-causing SARS-CoV-2 variant vs. against the D614G Reference strain. The dashed lines are pointwise 95% confidence intervals. The dots are VE point estimates against the given lineage, with the vertical bars showing 95% confidence intervals. The "double one-sided unadjusted sieve *p* value" doubles the *p* value from a one-sided Wald test of the null hypothesis of constant VE vs. the alternative hypothesis of a decreasing VE with an increasing value of the feature on the x-axis (Juraska and Gilbert[19], Section 5). CI confidence interval, nAb neutralizing antibody, Unadj. unadjusted.

associated with decreased VE after phylogenetic correction (FDR-adjusted *p* < 0.2). Because the GenSig tool runs a test at each AA position, to make sure it was indeed the full deletion to glycan mutation that yielded significance after phylogenetic correction, we reran after excluding 6 sequences with ambiguous calls in the deletion, and forced the 246–253 motifs into a single mutation. The N246 glycan change was significant after phylogenetic and multiple testing correction (FDR-adjusted *p* = 0.002). While no other AA position variables were significantly associated with VE after phylogenetic and multiple testing correction, other mutations characteristic of the Lambda variant (G75V, T76I, L452Q, F490S) and/or associated with changes in RBD class I and II bnAb binding (L452R, E484K, F490S) or neutralization potency (L18F, T20N, K417T/N, L452R) were significantly enriched in the breakthrough sequences from the vaccinated group (FDR-adjusted *p* < 0.10).

### Structure and immune evasion of the Lambda variant glycoprotein

To gain insight into the potential mechanisms underlying these observed Lambda variant sieve effects, we determined the structure of the Lambda-variant glycoprotein spike trimer bound to S309 (an RBD-

targeting nAb[35]), S2L20 (an NTD-targeting non-nAb[13]), and S2X303 (an NTD-supersite targeted nAb[36]) using cryoEM (Fig. 8a). S383C/D985C mutations[37] were used to staple the RBDs closed, thereby enabling the use of C3 symmetry to aid structural determination.

Local classification and refinement of the RBD bound to S309 improved the local resolution of this region, permitting atomic model building in the cryoEM density, including the L452Q mutation and the main chain of F490S (Fig. 8b). Both mutations at positions L452 and F490 are associated with escape from site Ib RBD-targeted antibodies. ELISA experiments confirmed that Lambda escapes three of a panel of 12 RBD-targeting antibodies (Fig. 8d). Yeast-display experiments showed that the F490S mutation does not improve binding to ACE2, while the L452Q mutation hardly has an effect on ACE2 binding[38]. Consistent with this study, binding experiments confirmed the Lambda RBD binds with approximately equivalent affinity to ACE2 as the index strain RBD (Fig. 8e).

Local classification and refinement of the NTD bound to S2L20 and S2X303 improved local resolution of this region, permitting model building, including the R246N mutation and the 247–253 deletion (commonly referred to as the "RSYLTPGD246-253N" mutation) (Fig. 8c). This deletion introduces a new glycan sequon and the linked glycan could be visualized and built in the cryoEM map. These mutations are in the NTD antigenic supersite loop and are therefore predicted to permit escape from NTD-targeted nAbs. ELISA experiments confirmed that Lambda escapes 10 of a panel of 11 NTD-targeted nAbs (Fig. 8d).

## Discussion

Sieve analysis compares genotype-specific or immunophenotype-specific COVID-19 incidence between randomized study groups, therefore directly assessing the causal effects of vaccination and providing inferences for how vaccine efficacy depends on SARS-CoV-2 features. In addition to the strength of a randomized, double-blinded, placebo-controlled phase 3 trial, the present sieve analysis of ENSEMBLE had ample statistical precision due to the large number of SARS-CoV-2 Spike sequences (measured from more than 1,200 participants) and the broad proteomic variability of the SARS-CoV-2 Spike sequences causing these endpoints. Consequently, the sieve analysis could provide many insights into how the efficacy of the Ad26.COV2.S vaccine, evaluated in baseline SARS-CoV-2 negative individuals, depended on virus features.

In the Latin American cohort, VE against the moderate to severe–critical COVID-19 primary endpoint significantly declined with Spike sequence distance as measured in myriad ways, including lineage, weighted Hamming distances calculated for Spike, RBD, NTD, and S1, scores reflecting degree of escape from epitope-specific antibodies computed using deep mutational scanning or based on crystal structures in the PDB, and NTD features previously shown to impact neutralization. Estimates of VE by lineage were consistently ordered by the distances of the different lineages to the vaccine strain. VE declined similarly with Spike, RBD, NTD, and S1 distances (VE about 70% against viruses closest to the vaccine and 20% against viruses beyond the 90–95th percentile of distances) but did not depend on S2 distances. This may be explained by S2's relative conservation when compared to S1. As such, almost all variant-characteristic mutations are not in S2, and none of the prescribed antibody-epitope footprint clusters included S2 positions (only rare epitopes in PDB mapped to S2), reflecting S2's stalk location and relative lack of exposure to the immune system.

VE significantly declined with 14 of the 20 evaluable antibody-epitope escape scores. Six antibody-epitope clusters had no evidence of impacting VE: DMS3, PDB2, PDB5, PDB6, PDB9, and PDB14. Of the 14 clusters with a sieve effect, nine include at least one site that harbors a characteristic mutation of Lambda, whereas three include site 417 which is a characteristic mutation of Mu and Gamma, one includes site 501 that harbors a characteristic mutation of Gamma, Alpha, and Mu,

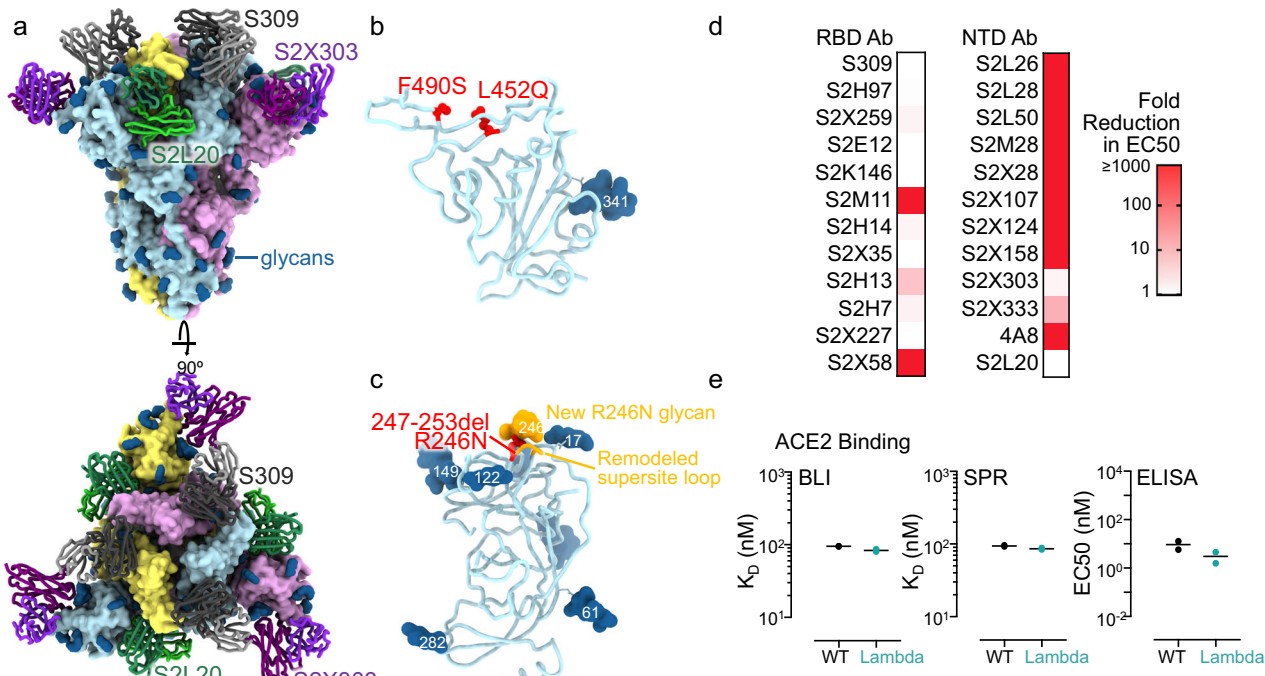

**Fig. 8 | CryoEM structure of the SARS-CoV-2 Lambda Spike ectodomain trimers and analysis of monoclonal antibody (mAb) and ACE2 binding. a** Structure of the Lambda Spike trimer (surface rendering) bound to the S2L20, S2X03, and S309 Fabs (ribbons). SARS-CoV-2 Spike protomers are colored pink, cyan, and gold, whereas the S2L20 Fab heavy and light chains are colored dark and light green, respectively. The S2X303 Fab heavy and light chains are colored dark and light purple, respectively. The S309 Fab heavy and light chains are colored dark and light orange, respectively. Only the Fab variable domains are resolved and therefore modeled in the map. N-linked glycans are rendered as dark blue spheres. **b** Zoomed-in view of the S309-bound Lambda RBD with L452Q and F490S shown as red spheres. **c** Zoomed-in view of the S2L20- and S2X303-bound Lambda NTD with the R246N mutation shown as red spheres; the remodeled loop caused by the 247–253 deletion as well as the new R246N glycan are shown in orange. **d** Binding of a panel of 12 neutralizing RBD-specific mAbs, 11 neutralizing (antigenic site i) NTD-specific mAbs, and one non-neutralizing (S2L20, antigenic site iv) NTD-specific

mAbs to recombinant SARS-CoV-2 Spike variants analyzed by ELISA displayed as a heat map (relative to SARS-CoV-2 index strain Spike binding). **e** (Left) Biolayer interferometry (BLI) binding analysis of the human ACE2 ectodomain (residues 1–615) to immobilized biotinylated SARS-CoV-2 index strain and Lambda RBDs. Data from one biological replicate is shown with 1–2 technical replicates each. (Center) Surface plasmon resonance (SPR) binding affinity analysis of the human ACE2 ectodomain (residues 1–615) for immobilized biotinylated SARS-CoV-2 index strain and Lambda RBDs. Data from two biological replicates are shown with two technical replicates each. (Right) ELISA binding analysis of the SARS-CoV-2 index strain and Lambda RBDs to immobilized human ACE2 ectodomain (residues 1–615 shown as 50% effective concentrations (EC50). Data from two biological replicates are shown with two technical replicates each. CryoEM cryo-electron microscopy, NTD N-terminal domain, RBD receptor-binding domain. Source data are provided as a Source Data file.

and one includes both sites 417 and 501. Thus the nine sieve-effect clusters appear to be driven by the differential VE by Lambda vs. not-Lambda, whereas the other five appear to be driven by mutations at the important sieve-effect sites 417 and 501 that impact neutralization. Of the six non-sieve-effect clusters, only one (PDB14) included a site harboring a characteristic mutation of Lambda, site 75, which was a sieve-effect site with FWER $p \leq 0.05$. The potential for sieve effects in different epitope sets depends on many factors, including the level of accessibility to nAbs, conservation, and the narrowness of the footprints on the tridimensional structure they target.

Given that most of the sieve effects found in this study are linked with Lambda, it is not surprising that prior work suggests that Lambda's mutations enable it to function as an immune escape variant. Kimura et al.[39], Wang et al.[40], and Acevedo et al.[24] showed that Lambda exhibited reduced neutralization sensitivity to antibodies induced by both vaccination and prior infection, with evidence suggesting that these effects were caused by the L452Q, F490S, and RSYLTPGD246-253N mutations. Kimura et al. and Acevedo et al. also considered other variants, including Gamma, and found that Lambda exhibited the least neutralization sensitivity among all those studied, and Kimura et al. found that Lambda was more infectious than the other variants due to its T76I and L452Q mutations. Additionally, ref. 41 directly compared Lambda with Delta and demonstrated that Lambda was more successful at evading vaccine-induced humoral

immunity, while showing similar binding affinity to ACE2 as the ancestral strain.

nAb assays have performed well at predicting vaccine efficacy against COVID-19 and severe-critical COVID-19 across SARS-CoV-2 lineages[15,16,42]. Importantly, one of the sieve analyses in the present work scored viruses by their lineage's directly measured resistance to neutralization by sera from ENSEMBLE Ad26.COV2.S vaccine recipients, providing a way to study a neutralization correlate of protection (CoP) in a complementary way to individual-level (e.g., refs. 43–46) and population-level immune correlates analyses (e.g., ref. 47). VE significantly declined against lineages with greater neutralization resistance scores, providing validation of pseudovirus neutralization titer as a CoP. However, while refs. 24,39–41. demonstrated that Lambda was more resistant to nAbs than other variants, Lambda was less resistant to neutralization by Ad26.COV2.S vaccinee sera than Mu, Gamma, and Zeta, such that the ordering of variants by this neutralization scoring was discordant with the level of vaccine efficacy against the variants, indicating that vaccine efficacy against Lambda was lower than predicted from a serum nAb-CoP model. While, to our knowledge, no studies have directly compared the immune response susceptibility of Lambda vs. Zeta and of Lambda vs. Mu, two of the studies above[24,39] demonstrated that Lambda showed a lower neutralization sensitivity than Gamma, suggesting that neutralization readouts from vaccinee sera cannot be used as the sole

basis of extrapolating predictions of VE, where other immune functions (e.g., cellular, Fc effector) could be important to include in such predictions. It is interesting to note that ref. 48 showed that Spike-specific CD4 + T-cell and CD8 + T-cell responses in Ad26.COV2.S vaccine recipients ($N$ = 28) were generally maintained across the Alpha, Beta, Gamma, Delta, B.1.1.519, Kappa, Lambda, and R.1 variants, although at the individual-donor level, one Ad26.COV2.S vaccine recipient showed reduced Lambda-Spike CD4 + T-cell responses and one showed reduced Lambda-Spike CD8 + T-cell responses (both vs. Ancestral-Spike responses). Additionally, the association of weighted Hamming distances with vaccine efficacy (Fig. 3) were concordant with vaccine efficacy by variant, especially for NTD, with Lambda being most distant from the vaccine strain (with notable outlying viruses with 25 residue mismatches to the vaccine strain) and against which vaccine efficacy was lowest. This suggests that amino acid sequence distances may have advantages as a biomarker for reliably predicting vaccine efficacy, a result also supported by ref. 7. A caveat of the neutralization sieve analysis is that the lineage scores were estimated from vaccinee sera from only eight ENSEMBLE participants, although the scores were supported by additional data from 17 Ad26.COV2.S vaccine recipients in the COV2001 phase 1/2a study[27].

The relative prevalence of SARS-CoV-2 lineages changed over time (Fig. 1a and Fig. 1 of ref. 6) where in Latin America the median (range) number of days from enrollment until the COVID-19 endpoint among placebo recipients was 48 (15, 197) for Reference.45, (15, 141) for Zeta, 114 (42, 220) for Gamma, 126 (57, 204) for Lambda, and 170 (109, 219) for Mu. If newer variants tended to expose participants later in follow-up than older variants, it could cause spurious genotypic sieve effects that are instead due to waning vaccine efficacy. This potential bias was mitigated by controlling for the calendar time of enrollment in the sieve analyses.

Given that the sieve effects identified in this analysis have strong linkages with viral variants (especially Lambda), a natural consideration is to identify which of these features are drivers of vaccine escape versus lineage-linked "bystander" features that are merely along for the ride. Among the four identified sieve sites in the RBD (414, 452, 490, 501), the deep mutational scanning work by ref. 9 identified sites 452 and 490 with significant antibody-escape in some variants, and that site 490 is in a Class Ib epitope targeted by antibodies such as LY-CoV555. Of the ten identified sieve sites in the NTD (75, 76, 246, 247, 248, 249, 250, 251, 252, and 253), eight of them fall in the NTD supersite identified by ref. 13, by way of Lambda's RSYLTPGD246-253N mutation. The end result of this mutation is an N-linked glycosylation motif starting at position 246, which is a novel motif only found in Lambda. The glycan at position 246 and the deletion remodel the supersite and promote NTD-targeted nAb escape, as confirmed by ELISA, showing that binding of 10 out of 11 tested NTD-targeted mAbs was abrogated. As R246A or R246Q point mutations in the Lambda spike abolish this glycan sequon and do not affect neutralization of BNT162b2 vaccine recipient sera[39], the 247–253 deletion rather than the new glycan may be solely responsible for the association of the RSYLTPGD246-253N mutation with decreased VE. These findings provide a potential mechanism by which the Lambda variant deletion RSYLTPGD246-253N in the NTD supersite epitope was associated with decreased VE after phylogenetic correction. The other two NTD sieve sites (75 and 76) are the trailing residues in an N-linked glycosylation motif beginning at position N74; similarly, as a glycan at this site may promote binding, mutations in these downstream positions disrupt the motif and may facilitate immune escape.

A phylogenetically corrected analysis of a subset of 1159 codon-aligned nucleotide Spike sequences confirmed a significantly lower VE against the Lambda variant. This analysis also found additional sites implicated in higher bnAb neutralization resistance to be enriched in the vaccine group breakthrough sequences. While these additional associations were not significant after the phylogenetic

correction, one needs to take into account the low diversity and recent evolutionary history of the Spike protein at the time of this analysis compared to HIV-1, for which the GenSig tool was originally designed, thus underpowering the overall analysis. Interestingly, many of these sites are enriched in the Lambda variant, against which VE was found to be significantly lower after phylogenetic correction. At the time it emerged, many studies indeed predicted higher resistance to RBD bnAbs, infectivity, and higher likelihood to escape vaccines in the Lambda variant compared to contemporary variants[25,41].

Another important consideration in the relevance of these results is how they are reflected in the current epidemic. The Ad26.COV2.S vaccine sieve effects observed here, based on data collected prior to July 10, 2021, revealed broader vaccine adaptation features as several sieve signature sites showed mutations in subsequent variant waves, including many that are still circulating today. At the time of this writing (October 2023), mutations at sites 252, 484, 490, and 501 are dominant in currently circulating Omicron sub-lineages and recombinants, including BA.2.86, EG.5.1, FL.1.5.1, GL.1, and HK.3. (Global proportion between June 27, 2023 and September 24, 2023: G252V = 75.2%; E484A = 96.5%; F490S = 93.2%; N501Y = 96.9%[49].) Of note is the sieve signature site F490S. While rare until the end of 2022, this mutation became dominant in early 2023 with the rapid global spread of the XBB.1.5 recombinant lineage[50].

Other mutations correlating with vaccine efficacy in this analysis have shown to be characteristic of other transient lineages, although they did not persist as long-term circulating features: e.g., G75V with DZ.2 and XBB.1.14; T76I with BA.5.5; S247N with BF.31.1; Y248D with BQ.1.1.45; T250I with AY.33; and P251H with BQ.1. L452R exhibited a limited show of prominence up through the start of 2022 with the Delta variant AY.4, and again in the latter half of 2022 with the Omicron sub-lineage BA.5.2.1, but has since faded from circulation. Even Q414R and T859N, the two VE-associated mutations that were not characteristic of any of the WHO-labeled variants observed in this study, emerged briefly as characteristic mutations in future minor lineages (AY.29.2 and B.1.637, respectively).

The fact that sieve analysis predicted currently relevant mutations could be expected, since SARS-CoV-2 has shown remarkable patterns of convergent evolution since the initial appearance of variants, with numerous recurrent mutations, especially in the RBD, shared across lineages over time[51]. Conversely, some of the sites found to be associated with Ad26.COV2.S efficacy in this analysis (R246 and L249) were not strongly characteristic of any future lineages of note. The R246N mutation in Lambda, as mentioned above, results in a novel N-linked glycosylation motif, and its apparent absence in future lineages seemingly validates Kimura et al.'s finding that a glycan at this position does not impact the virus's fitness. Site 249, while also a part of the NTD supersite, seemingly does not tolerate substitutions very well, as its wild-type residue (L) is currently present in over 99% of Spike sequences, with the next-most prevalent mutation being a deletion. As such, its identification as a sieve site in this analysis may be due to the happenstance of being part of Lambda's RSYLTPGD246-253N mutation.

A strength of this study was that it was conducted in three separate geographic regions with different circulating lineages, which contribute insights based on these lineages and their characteristic signature mutations, and different distributions of genetic distances of circulating sequences to the vaccine strain. The analyses of Latin American study sites provided the greatest insights given that 63% of primary COVID-19 endpoints with sequence data were in Latin America, where the circulating SARS-CoV-2 sequences were the most diversified. All features showing sieve effects in the US also showed sieve effects in Latin America, constituting independent replication of results. The result of no sieve effects in South African study sites can likely be explained by the vast majority of circulating sequences being

Beta or Delta variants with limited dynamic range of genetic distances within each variant and a lack of Reference viruses that are close to the vaccine strain.

Another strength of this study was that VE against severe-critical COVID-19 could be assessed. The results support that VE against this endpoint also declines with Spike sequence distance as measured in multiple ways, yet with VE starting higher against viruses closest to the vaccine strain and diminishing less rapidly with increasing degrees of sequence mismatch. Thus, the results generally endorse the hypothesis that a single dose of Ad26.COV2.S vaccine will durably protect against severe outcomes. However, it is unlikely to protect against reinfection, as might be expected of any member of the coronavirus family[52]. It is difficult to deconvolute the effects of antigenic change on protection from severe outcomes and protection from reinfection from the natural decay of infection-blocking immunity as compared to the near permanence of immunity against severe outcomes.

Overall, the finding that protection against severe-critical COVID-19 is more invariant to sequence changes than against less-symptomatic COVID-19 may have clinical implications for planning updates of vaccines with new variants. The severe-critical classification covers a broad spectrum of clinical phenotypes ranging from individuals with only repeated low partial pressure of oxygen to severe pneumonia requiring respiratory support. Protection against hospitalization with severe consequences is clinically most important but sieve analysis specific to this outcome could not be performed given the small number of cases. Yet, ENSEMBLE and post-approval trials have shown high Ad26.COV2.S efficacy against this outcome, especially in South Africa after a 6-month boost, suggesting that neutralization resistance and sequence variation may be playing a less-dominant role in vaccine-induced protection against the most serious disease, perhaps due to CD8 + T cells[53].

## Methods

### Trial design, study cohort, and COVID-19 endpoints

Trial enrollment began on September 21, 2020. Participants were not compensated for their participation. The end of the double-blind period varied by country; the data cutoff for this analysis was July 9, 2021. The main endpoint for sieve analysis is the same COVID-19 primary endpoint (moderate to severe–critical) as in the primary analyses[6,17], restricting to endpoints starting 14 days post-vaccination. Moderate COVID-19 was defined by a positive RT-PCR test for SARS-CoV-2 as well as two or more of the following symptoms (new or worsening): fever or chills, cough, heart rate ≥90 beats/minute, muscle or body pain, headache, new loss of taste or smell, sore throat, red or bruised-looking feet or toes, nausea, vomiting, or diarrhea; or one or more of the following signs or symptoms: shortness of breath, respiratory rate >20 breaths/minute, clinical or radiologic evidence of pneumonia, deep vein thrombosis, or abnormal oxygen saturation (but above 93%)[6,17].

Severe–critical COVID-19 was defined by a positive RT-PCR test for SARS-CoV-2 with one of the following features: respiratory failure; evidence of shock (systolic blood pressure <90 mm Hg, diastolic blood pressure <60 mm Hg, or requiring vasopressors); respiratory rate >30 breaths/min; heart rate ≥125 beats/min; oxygen saturation of 93% or less (ambient air at sea level), or a ratio of the partial pressure of oxygen to the fraction of inspired oxygen <300 mm Hg; intensive care unit admission; significant acute renal, hepatic, or neurologic dysfunction, or death[6,17].

Sieve analyses were also conducted for severe–critical COVID-19, again using the same definition as used in the primary papers[6,17]. Analyses were conducted in the per-protocol baseline seronegative cohort[17]. See Section 1 of the SAP (provided in ref. 20 and at the end of the Supplementary Information) and Supplementary Methods for further details.

### SARS-CoV-2 sequencing and sequence data

The Virology Laboratory at the University of Washington, Department of Laboratory Medicine and Pathology ("UW Virology") conducted next-generation sequencing of SARS-CoV-2 Spike sequences with the Swift Biosciences SNAP workflow version 2.0 on Illumina platforms[17,54]. Only Spike gene sequence information was obtained, and the assignment of WHO-labeled variants was based on profiles of predefined and characteristic amino acid substitutions in the Spike protein relative to the reference sequence [GenBank accession number NC_045512 (https://www.ncbi.nlm.nih.gov/nuccore/1798174254)]. For SARS-CoV-2 sequences, Nextclade (https://clades.nextstrain.org/)[55] and Pangolin (https://cov-lineages.org/resources/pangolin.html)[56] were used for lineage assignments. Sequenced samples were obtained as close as possible to the start of the symptoms from individuals acquiring a COVID-19 primary endpoint, where typically but not exclusively, samples with SARS-CoV-2 viral load above 1,000 copies/mL were shipped for sequencing.

When the sequencing was successful, the consensus sequence from the set of sequencing reads from a single run on a single clinical sample was used as the sequence in the analysis. For the 109 participants with sequences from more than one sample (97 with sequences from two samples and 12 with sequences from three samples), the consensus sequence from the chronologically earliest timepoint was used in the analysis.

Sequences were selected for analysis if they were obtained within 36 days following the first RNA-positive timepoint associated with the first moderate to severe-critical COVID-19 primary endpoint. See Supplementary Methods for further details.

In our analyses, we compared the observed sequences with the sequence to the insert of the Ad26.COV2.S vaccine. The vaccine insert strain/sequence is from the index strain (prior to the D614G mutation) with two stabilizing mutations at K986P and V987P, and two additional mutations (R682S and R685G) in the furin cleavage site. All site positions mentioned are relative to those in the index sequence.

### nAb titers

nAb titers were measured to a panel of Spike antigens representing the Reference strain B.1.D614G and several variants[27,28]. The nAb assay used a commercial cell line consisting of HEK293T target cells stably expressing the human ACE2 and human TMPRSS2 genes [CoronaAssay-293T(hACE2-hTMPRSS2), procured from VectorBuilder; Cat. CL0015]. R (version 3.4.3) was used to calculate SARS-CoV-2 neutralizing titers using a four-parameter curve fit as the sample dilution at which a 50% reduction (IC50) of luciferase readout was observed compared with luciferase readout in the absence of serum (High Control). Relative light units were measured using an EnSight Multimode Plate Reader (Perkin Elmer) running Kaleido software (version Kaleido 3.0.3067.117x). Each variant was assigned a score defined as the log10-transformed ratio of the geometric mean titer of vaccinee sera against the variant and the geometric mean titer of vaccinee sera against the Reference strain.

### Sieve analysis

This analysis was prespecified and documented in the SAP. The sequences and clinical data were pre-processed into an analysis dataset as specified by the SAP, using R[57] (version 4.3.1) with the seqinr[58] package (version 4.2−30). The sieve analyses were conducted for each of the four geographic regions: Latin America, South Africa, the United States, and the three geographic regions pooled (hereafter, "geographic-region analyses"). For each geographic-region analysis, lineages with at least 20 COVID-19 endpoints were included. For amino acid (AA) position scanning sieve analysis that considered residue match-vs.-mismatch to the vaccine-strain residue, positions with at least 20 COVID-19 endpoints with a residue match and at least 20 COVID-19 endpoints

with a residue (or gap/insert/deletion) mismatching the vaccine-strain residue, were included. Similar AA position scanning sieve analyses were done that focused on specific residues at given positions, where residues at positions with at least 20 COVID-19 endpoints with the residue and at least 20 COVID-19 endpoints without the residue, were included. The same screening rule was used for the Latin America country-specific analyses. For the severe-critical COVID-19 endpoint, ten severe-critical COVID-19 endpoints were used to down-select the lineages and AA position features for each geographic-region analysis. Further details are given in Section 1.6 of the SAP.

While the primary analyses[6,17] both counted endpoints starting 14 or 28 days post-vaccination, all of the sieve analyses restrict to starting 14 days post-vaccination, given that similar results are expected and more COVID-19 endpoints could be included in the analysis.

## Specification of Spike AA sequence features for sieve analysis
We performed unsupervised learning of the treatment-blinded trial sequence data to fully specify and down-select the set of AA sequence features that were studied for sieve effects. All statistical inferences were prespecified in the SAP before treatment unblinding. Then, inferential statistical analysis (supervised learning that produces VE estimates, differential VE estimates, confidence intervals, and $p$ values) was conducted in an automated/press-button fashion, with the inferences valid based on the pre-specification of inferences and the reproducibility of the computer code.

Features were classified into two types: (1) all Spike AA sequence features with sufficient variability to study for potential sieve effects ("All" features) and (2) the subset of All features

that are directly connected to a hypothesis that nAbs are a correlate of protection, and were selected based on knowledge/data or hypotheses that different levels of the feature affect neutralization (nAb-CoP-hypothesis features). See Section 1.3 of the SAP for further details.

## Handling of missing sequences
For primary endpoint COVID-19 cases, either all Spike sequence features are observed, or no Spike sequence features are observed. Thus, the structure of the missing data pattern for primary endpoint cases is simple, with complete sequence data or no sequence data. See Section 2.4 of the SAP for details on how each specific sieve analysis method handles the missing sequence data.

## Quantification of viral diversity
Spike sequence diversity within each geographical location (Latin America, South Africa, United States of America) was quantified using Rao's $Q$, which measures the average phylogenetic distance between any two Spike AA sequences randomly chosen from any given region[59].

## Structural modeling
All protein structures were generated using VMD[60] (version 1.9.4). PDB files were obtained from the RSCB Protein Data Bank (PDB) and are specified as described in Supplementary Methods. Using Python (version 3.9), NumPy[61] (version 1.20.3) was used for data processing and MDAnalysis[62,63] (version 2.0.0) was used for processing, editing, and generating PDB structure files. All protein backbones were drawn in the NewCartoon style to keep focus on the residues and regions of interest. In order to apply a weighted color to the residues, a Python script (publicly available at ref. 64) was used to translate cluster weights into a VMD readable scale metric. This metric, the beta factor, can be used as a dummy field in PDB files and applies a red-to-blue gradient at values of 0 and 200, respectively. Cluster weights were set to a baseline of 100 before adding the cluster weight scaled by a factor of 20 in order to highlight cluster variation.

## AA sequence sieve analysis methods: prospective VE sieve analysis
For sieve analyses that answer the questions of whether and how VE depends on AA sequence features of exposing SARS-CoV-2 viruses [prospective VE sieve analysis[65]], the Spike AA sequence feature-specific VE estimands for measuring sieve effects are defined in the SAP. The main estimand used is genotype-specific hazard-ratio VE, which for a given genotype is defined as 100% times one minus the genotype-specific hazard ratio (vaccine/placebo) of COVID-19 over the follow-up period 14 days post-vaccine or placebo administration until unblinding. An AA sequence sieve effect is defined as statistically significant evidence for differential VE across multiple levels of a given AA sequence feature. Details on the sieve analysis methods are given in Section 2 of the SAP.

For these hazard ratio-based prospective VE sieve analyses, the following software was used: R 4.2.3[57] and Rstudio[66] 2023.03.0 + 386. For hazard ratio-based sieve model fitting, the R packages sievePH[19,67] (version 1.0.4) and cmprskPH[68] were used. Information on additional R packages used is provided in the Supplementary Methods.

## Multiple hypothesis testing adjustment for AA sequence sieve analysis
For the AA sequence sieve analysis, the following plan was implemented for multiple testing adjustment, separately for each geographic region-specific analysis:

Family-wise error rate (FWER) adjusted $p$ values are Holm–Bonferroni and FDR-adjusted $p$ values ($q$ values) are Benjamini–Hochberg, computed separately for the two per-protocol baseline seronegative cohort analyses defined by All features and nAb-CoP-hypothesis features, and within each of these analysis types separately for each of the classes of defined sequence feature sets given in Section 2.11 of the SAP. Significant results are marked at two levels of evidence, the higher evidence being FWER $p \le 0.05$ and the lesser evidence being all three outcomes of unadjusted $p$ value $\le 0.05$, $q$ value $\le 0.20$, and FWER $p > 0.05$. Results in the text with $q$ value $\le 0.20$ are only reported as such if also the unadjusted $p$ value $\le 0.05$.

## Classification sieve analysis
For the classification sieve analyses assessing multivariable viral predictors of the treatment arm, we estimated multivariable prediction functions using a Super Learner ensemble[69] and performed a variable importance analysis, defining variable importance as the difference in area under the receiver operating characteristic curve between including versus excluding a group of virus features in COVID-19 endpoint cases. We computed point estimates and $p$ values for each group. R[57] (version 4.3.1) was used. Information on the R packages used is provided in the Supplementary Methods.

## Deep mutational scanning (DMS) antibody-escape scores
Various antibody-escape scores representing how mutations in RBD impact antibody binding[26] (and hence likely relevant to neutralization) were calculated. This involved two steps: identification of epitope-specific escape scores based on putative antibody-epitope footprints, and subsequent calculation of the antibody-escape scores for each sequence. These steps are detailed in Supplementary Methods.

## Protein Data Bank (PDB) antibody-escape scores
The PDB antibody-escape scores are based on sets of putative antibody footprint sites, which are associated with Spike-antibody structural interactions. These site sets were determined from unique SARS-CoV-2 Spike and human anti-Spike antibody complexes that were downloaded from the PDB database (https://www.rcsb.org/) ($n = 274$ on May 4, 2022). For each PDB complex, epitope sites were defined as antigen sites that are in contact with the antibody in the antigen-antibody

complex (i.e., all sites that have non-hydrogen atoms within 4 Angstrom of the antibody). Quantitation of the interaction between an epitope site and the antibody is detailed in Section 2.1.5 of the SAP and Supplementary Methods.

## Neutralization hypothesis-driven sieve analysis

Mutations, substitutions, insertions and/or deletions at the following locations were identified as impacting neutralization in in vitro experiments: residues at AA positions 14–20, 140–158, and 245–264 that encompass the supersite epitope; mutations in the NTD signal peptide that may impact where the signal peptide is cleaved off (e.g., S12P or S13I); residues at positions 12-13, 14–26, 138–158, 242–264; D80 mutations; deletions at positions 69 and 70; mutation T95I; and R190 mutations. Based on these observations, we defined the following seven dichotomous NTD features:

• NTD1: One or more deletions or insertions in positions 138–158 and 242–264;

• NTD2: Mutations at positions 12 or 13, which may delay cleavage of the signal peptide;

• NTD3: One or more of the following mutations, R246X, G142X, K147X, or L18P;

• NTD4: One or more substitutions in 138–158 excluding positions 142 and 147;

• NTD5: One or more substitutions in 14–26 and 242–264 excluding substitutions R246X and L18P;

• NTD6: One or more substitutions D80X or L18F;

• NTD7: One or more substitutions that add or remove a glycan sequon within the NTD.

Further details are given in Section 2.1.6 of the SAP.

## Covariability analysis

The covariability of two AA positions is quantified by the normalized mutational information Mstar, which has a range from 0 to 1[21]. Mstar equals the likelihood ratio statistic for testing the independence of the AA changes at the two AA positions, normalized to weight the positions equally regardless of diversity. For calculating the covariability of an AA position with a lineage, Mstar is calculated in the same way as for a pair of AA positions, using two levels of the lineage variable present vs. absent.

## Phylogenetic trees

The overall phylogeny was reconstructed by approximate maximum likelihood with FastTree v2.1.11 compiled with double precision[70], under the Jones–Taylor–Thorton (JTT) substitution model and CAT approximation with 20 rate categories[71]. This tree was rooted by the Reference group, and the sequences for the vaccine insert and the reference sequence [GenBank accession number NC_045512 (https://www.ncbi.nlm.nih.gov/nuccore/1798174254)] were included for comparison. The trees for each region were extracted and then visualized using the ggtree[72] and patchwork[73] packages in R[57] (version 4.1.1.4).

## GenSig

We used the LANL tool GenSig to identify signature sites associated with vaccine status after a phylogenetic correction that accounts for potentially spurious associations due to lineage effects[30,31]. Further details are in the Supplementary Methods.

## Production of recombinant glycoproteins

The SARS-CoV-2 S Lambda ectodomain contains Lambda mutations G75V, T76I, R246N, 247-253del, L452Q, F490S, D614G, and T859N, in addition to Hexapro stabilizing mutations[74], and DS RBD stapling mutations[37]. S trimer was produced in 100 mL cultures of Expi293F Cells (Thermo Fisher Scientific, #A14527) grown in suspension using Expi293 Expression Medium (Thermo Fisher Scientific) at 37 °C in a humidified 8% $CO_2$ incubator rotating at 130 rpm. Cells grown to a density of 3 million cells per mL were transfected using the Expi-Fectamine 293 Transfection Kit (Thermo Fisher Scientific) and cultivated for 4 days, at which point the supernatant was harvested. S ectodomains were purified from clarified supernatants using a Cobalt affinity column (Cytiva, HiTrap TALON crude), washing with 20 column volumes of 20 mM Tris-HCl pH 8.0 and 150 mM NaCl, and eluted with 20 mM Tris-HCl pH 8.0, 150 mM NaCl, and 600 mM imidazole. The S ectodomain was then concentrated using a 100 kDa centrifugal filter (Amicon Ultra 0.5 mL centrifugal filters, MilliporeSigma), residual imidazole was washed away by consecutive dilutions in the centrifugal filter unit with 20 mM Tris-HCl pH 8.0 and 150 mM NaCl, and finally concentrated to 1 mg/mL before use immediately after purification. The RBD construct was based on reported constructs (with the exception of Lambda L452Q and F490S mutations in the RBD) and were produced and biotinylated as previously[75]. The hACE2 construct was synthesized by GenScript into pCMV (residues 19–615 from Uni-Prot Q9BYF1 with a C-terminal AviTag-10xHis-GGG-tag, and N-terminal signal peptide) and was expressed in HEK293.sus using standard methods (ATUM Bio). The hACE2 protein was purified via Ni Sepharose resin followed by isolation of the monomeric hACE2 by size exclusion chromatography using a Superdex 200 Increase 10/300 GL column pre-equilibrated with PBS.

## CryoEM sample preparation and data collection

Fabs were generated by LysC digestion [1:3000 (w/w) antibody:LysC] for 16 h at 37 °C. Next, 50 μL of 2 mg/mL SARS-CoV-2 S Lambda ectodomain was incubated with 40 μl 3.4 mg/mL S309 Fab, 3.6 μl 28 mg/ml S2X303 Fab, and 2.2 μL of 67 mg/mL S2L20 Fab in 150 mM NaCl and 20 mM Tris-HCl pH 8 for 15 min at 37 °C. Unbound Fab was then washed away with six consecutive dilutions in 400 μL of 20 mM Tris-HCl pH 8.0 and 150 mM NaCl over a 100 kDa centrifugal filter (Amicon Ultra 0.5 mL centrifugal filters, MilliporeSigma). The complex was concentrated to 3.5 mg/mL and 3 μL was immediately applied onto a freshly glow discharged 2.0/2.0 UltraFoil grid (84) (200 mesh), plunge frozen using a Vitrobot Mark IV (Thermo Fisher Scientific) using a blot force of −1 and 6.0 s blot time at 100% humidity and 23 °C. Data were acquired using the Leginon software[76] to control a FEI Titan Krios transmission electron microscope equipped with a Gatan K3 direct detector and operated at 300 kV with a Gatan Quantum GIF energy filter. The dose rate was adjusted to 3.75 counts/super-resolution pixel/s, and each movie was acquired in 75 frames of 40 ms with a pixel size of 0.843 Å and a defocus range comprised between −0.4 and −2.0 μm.

## CryoEM data processing

Movie frame alignment, estimation of the microscope contrast-transfer function parameters, particle picking, and extraction (with a down-sampled pixel size of 1.686 Å and box size of 256 pixels²) were carried out using Warp[77]. Reference-free 2D classification was performed using cryoSPARC[78] to select well-defined particle images. 3D classification with 50 iterations each (angular sampling 7.5° for 25 iterations and 1.8° with local search for 25 iterations) were carried out using Relion without imposing symmetry. 3D refinements were carried out using non-uniform refinement in cryoSPARC (62) before particle images were subjected to Bayesian polishing using Relion[79], during which particles were re-extracted with a box size of 512 Å at a pixel size of 0.843 Å. Next, 86 optics groups were defined based on the beam tilt angle used for data collection. Another round of non-uniform refinement in cryoSPARC was then performed concurrently with global and per-particle defocus refinement. For focused classification of the NTD and RBD, particles were symmetry-expanded and 3D classified in Relion without alignment, and then particles in well-formed 3D classes were then used for local refinement in cryoSPARC. For the NTD, the mask that encompasses the NTD, the S2L20 VH/VL region, and the S2X303 VH/VL region. For the RBD, the mask encompasses the RBD and the S309 VH/VL region. Reported resolutions are based on the

gold-standard Fourier shell correlation of 0.143 criterion and Fourier shell correlation curves were corrected for the effects of soft masking by high-resolution noise substitution[80,81].

Supplementary Table 28 provides information on cryoEM data collection, refinement, and validation statistics.

### CryoEM model building and analysis
UCSF Chimera[82] and Coot[83] were used to fit atomic models of S2L20, S309, and SARS-CoV-2 S (PDB 7SOB) into the cryoEM maps. The model was then refined and rebuilt into the map using Coot, Rosetta[84,85], and ISOLDE[86]. Model validation and analysis used Phenix[87]. Figures were generated using UCSF ChimeraX[88].

### ACE2 binding measurements using Biolayer interferometry
Lambda RBD was biotinylated and immobilized at 5 ng/μL in undiluted 10X kinetics buffer (Pall) to SA sensors that were pre-hydrated in water for 10 min and then equilibrated into 10X Kinetics Buffer (Pall). The RBDs were loaded to a level of 1 nm total shift. The loaded tips were then dipped into a dilution series of monomeric ACE2-his in 10X Kinetics Buffer (Pall) starting at 1000 or 5000 nM for 300 s prior to 300 s dissociation in 10X Kinetics buffer for kinetics determination. The data were baseline subtracted, and the plots were fitted using the Pall FortéBio/Sartorius analysis software (v.12.0). Data were plotted in GraphPad Prism (v.9.0.2).

### ACE2 binding measurements using surface plasmon resonance
RBD:ACE2 affinity measurements were performed using a Biacore T200 instrument. The Cytiva Biotin CAPture Kit, Series S, was used for surface capture of biotinylated RBD. The running buffer was HBS-EP+ pH 7.4 (Cytiva) and measurements were performed at 25 °C. Experiments were performed with a threefold dilution series of monomeric hACE2: 300, 100, 33, 11, 3.7 nM. The association was 300 s and dissociation was 450 s. Data were double reference-subtracted and fit to a 1:1 binding model using Biacore Evaluation software.

### ELISA
For ELISA experiments with NTD-targeted mAbs, 384-well Maxisorp plates (Thermo Fisher Scientific) were coated overnight at 4 °C with 2 μg/mL of S glycoprotein in 20 mM HEPES pH 8 and 150 mM NaCl. For ELISA experiments with RBD-targeted mAbs, 384-well Maxisorp plates (Thermo Fisher Scientific) were coated overnight at 4 °C with 4 μg/mL of hACE2-His in 20 mM Sodium Phosphate pH 8 and 100 mM NaCl. The antibodies that were used were purified previously[13,36]. Plates were slapped dry and blocked with Blocker Casein in TBS (Thermo Fisher Scientific 37532) for one hour at 37 °C. Plates were slapped dry and mAbs were serially diluted in TBST with an initial concentration of 50 μg/ml. Plates were left for one hour at 37 °C and washed 4X with TBST, then 1:5000 Goat anti-Human (Thermo Fisher Scientific A18817) was added. Plates were left for 1 h at 37 °C and washed 4x with TBST, and then TMB Microwell Peroxidase (Seracare 5120-0083) was added. The reaction was quenched after 4 min with 1 N HCl and the A450 of each well was read using a BioTek plate reader.

### Sex and gender in reporting
Information on participant sex was self-reported, solicited, and collected by four predefined options (female, male, unknown, and intersex).

Sadoff et al.[17] determined that sex had no meaningful impact on vaccine efficacy. As such, we scoped our sieve analyses accordingly, as any finding of viral features impacting vaccine efficacy by sex would likely be a false discovery or would need to be interpreted as a qualitative interaction.

### Inclusion and ethics
The COV3001 (ENSEMBLE) study was reviewed and approved by all relevant local ethics committees and Institutional Review Boards. All participants provided written informed consent. All experiments were performed in accordance with the relevant guidelines and regulations.

Site PIs were invited as co-authors according to the enrollments performed in the study, and were given the opportunity for intellectual contribution.

### Reporting summary
Further information on research design is available in the Nature Portfolio Reporting Summary linked to this article.

## Data availability
The sequence data used in this study are available in two groups: Information pertaining to the SARS-CoV-2 sequences obtained from study participants, including their GISAID accession numbers, is provided in the Supplementary Data 1 file. The sequences curated by LANL to define the canonical variant sequences are available on GISAID through identifier EPI_SET_221208yn (https://doi.org/10.55876/gis8.221208yn)[89]. Available information includes contributors' details, such as accession number, virus name, collection date, originating lab, submitting lab, and the list of authors.

The deep mutational scanning (DMS) data used to identify the DMS antibody-escape scores are available at https://raw.githubusercontent.com/jbloomlab/SARS2_RBD_Ab_escape_maps/651fe6fa5a7fccec2b662ddbb45b6d2c7421ae74/processed_data/escape_calculator_data.csv. The representative Protein Data Bank (PDB) complexes for the PDB escape scores (Supplementary Table 2) are available from the PDB (https://www.rcsb.org/).

The cryoEM structures have been deposited at the PDB (https://www.rcsb.org/) and at the EMDB under the following accession numbers: D_1000281320 = Lambda global refinement, PDB: 8VYE, EMDB: EMD-43658; D_1000281321 = Lambda NTD local refinement, PDB: 8VYF, EMDB: EMD-43659; D_1000281322 = Lambda RBD local refinement, PDB: 8VYG, EMDB: EMD-43660.

Source data for Fig. 8 and Supplementary Figs. 53, 54 are provided with this paper.

The data sharing policy of Janssen Pharmaceutical Companies of Johnson & Johnson is available at https://www.janssen.com/clinical-trials/transparency. The data needed to execute the custom code for the sieve analysis as well as the neutralizing antibody data supporting the findings of this study, are proprietary to Janssen and may be obtained from the authors upon reasonable request as determined by an agreement with Yale Open Data Access (YODA) Project to serve as the independent review panel for evaluation of data requests. Project metrics for past data requests via YODA are available at https://yoda.yale.edu/metrics/. Source data are provided with this paper.

## Code availability
All custom code for the sieve analysis, including code for: the unsupervised learning of the treatment-blinded trial sequence data to fully specify and down-select the set of amino acid (AA) sequence features that were studied for sieve effects, implementing hazard-based sieve analysis, covariability analysis of any pairs of AA positions, the SuperLearner-based supervised learning sieve analysis, reproducing the figures in the supplemental material, calculating the epitope distance analyses, and generating the structural visualizations in the manuscript is publicly available at Figshare (https://doi.org/10.6084/m9.figshare.24911373.v1)[64].

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

## Acknowledgements

This research was supported in part by the Administration for Strategic Preparedness and Response, Biomedical Advanced Research and Development Authority, Government Contract Nos. HHSO100201700018C with Janssen; the National Institute of Allergy and Infectious Diseases (NIAID) grant UM1 AI068635 (HVTN SDMC) (PBG), UM1 AI068614 (HVTN LOC) (LC), and R37AI054165 (PBG); the Intramural Research Program of the NIAID Scientific Computing Infrastructure at Fred Hutch, ORIP grant S10OD028685; a cooperative agreement between The Henry M. Jackson Foundation for the Advancement of Military Medicine, Inc., and the U.S. Department of the

Army [W81XWH-18-2-0040] (MR, BLD, and HB); and by Janssen Research and Development, an affiliate of Janssen Vaccines and Prevention and part of the Janssen pharmaceutical companies of Johnson & Johnson. This study was also supported in part by the National Institute of Allergy and Infectious Diseases (P01AI167966, DP1AI158186 and 75N93022C00036 to DV), a Pew Biomedical Scholars Award (DV), an Investigators in the Pathogenesis of Infectious Disease Awards from the Burroughs Wellcome Fund (DV), the University of Washington Arnold and Mabel Beckman cryoEM center and the National Institute of Health grant S10OD032290 (to DV). DV is an Investigator of the Howard Hughes Medical Institute and the Hans Neurath Endowed Chair in Biochemistry at the University of Washington. The content is solely the responsibility of the authors and does not necessarily represent the official views of the National Institutes of Health. The findings and conclusions in this report are those of the author(s) and do not necessarily represent the views of the Department of Health and Human Services or its components. The views expressed by MR, HB, and BLD are those of the authors and should not be construed to represent the positions of the U.S. Army, the Department of Defense, or the Department of Health and Human Services. We thank Jesse Bloom for their input in defining deep mutational scanning Spike sequence features for sieve analysis. We also thank Davide Corti for the antibodies and SPR analysis. We gratefully acknowledge all data contributors, i.e., the Authors and their Originating laboratories responsible for obtaining the specimens, and their Submitting laboratories for generating the genetic sequence and metadata and sharing via the GISAID Initiative, on which this research is based. Final thanks go to Chenchen Yu for creating the tables of study participants' demographic information. Role of the sponsor: Janssen contributed to the study design, clinical data collection, and laboratory data collection and analysis (nAb assay); Janssen co-authors revised and approved the final version of the manuscript; and the manuscript passed through a formal internal review at Janssen. Otherwise, the study sponsors had no role in the immune correlates study design, immunogenicity data collection, and analysis, or manuscript writing.

## Author contributions

Conceptualization: CAM, AdC, MR, SR, AV, DJS, MLG, DF, PBG; Methodology: CAM, LL, AdC, MR, MJur, BDW, JL, CM, DB, AL, BS, FH, YS, AG, PR, OH, PBG; Software: CAM, LL, AdC, MR, MJur, BDW, JL, CM, DB, BS, FH, BLD, AG, PR, OH, PBG; Validation: CAM, LL, AdC, MR, MJur, BDW, JL, CM, DB, BS, FH, YS, AG, PR, OH, PBG; Formal Analysis: CAM, LL, AdC, EEG, MJon, BB, MMcC, MR, MJur, BDW, JL, CM, DB, AL, BS, FH, HB, AG, PR, OH, PBG; Investigation: MJon, BB, JS, GEG, MMcC, JEB, BG, SDR, DJS, MLG, JV, PAG, CT, IVD, ES, MAM, KMN, LC; Resources: JS, GEG, SDR, AV, DJS, MLG, JV, BG, PAG, LPdS, MSTS, MC, MHL, SJL, AG, LGB, NG, CT, IVD, ES, MAM, KMN, LC; Data Curation: CAM, LL, AdC, MR, MJur, BDW, JL, CM, DB, AL, AG, PR, CT, OH, PBG; Visualization: CAM, LL, AdC, MR, MJur, BDW, JL, CM, DB, AL, LNC, AG, PR, OH, PBG; CryoEM preparation, data collection, processing, model building, and model refinement: MMcC. Protein expression and purification: MMcC, JEB. Binding experiments: JEB, MMcC. Funding acquisition: LC, PBG, MR; Project administration: PBG; Supervision: DV, AG, PR, PBG; Writing – original draft: CAM, AdC, MR, LNC, PBG; Writing—review and editing: All.

## Competing interests

ALG reports contract testing from Abbott, Cepheid, Novavax, Pfizer, Janssen, and Hologic and research support from Gilead and Merck. JS declares support for the submitted work from the Janssen Pharmaceutical Companies of Johnson & Johnson and partial support (in the form of funding to his institution) from BARDA for the submitted work, declares support within the past 36 months from the Janssen Pharmaceutical Companies of Johnson & Johnson and BARDA funding for part of this work, has patents (US 11,384,122 B2) on invention of the Janssen COVID-19 vaccine, and has Johnson & Johnson stock and stock options. SR had partial support from the Department of Health and Human Services BARDA (in the form of contract payments to his institution) for the submitted work, has stock and/or stock options in Johnson & Johnson, and is an employee of Janssen Pharmaceutica NV. AV had partial support from BARDA (in the form of contract payments to her institution) for the submitted work, had all patent rights (US 11,384,122 B2) transferred to Johnson & Johnson, has stock and/or stock options in Johnson & Johnson, and is an employee of Janssen Pharmaceutica NV. DJS had partial support from the Department of Health and Human Services BARDA (in the form of contract payments to his institution) for the submitted work, has stock and/or stock options in Johnson & Johnson, and is an employee of Janssen. MLG had partial support from BARDA (in the form of contract payments to his institution) for the submitted work, has patents (US 11,384,122 B2) on the invention of the Janssen COVID-19 vaccine, has shares in Johnson & Johnson, and is an employee of Johnson & Johnson. JV has stock and stock options in Johnson and Johnson and is an employee of Janssen Pharmaceutica NV. CT and IVD both had partial support from BARDA (in the form of contract payments to their institution) for the submitted work, hold stock in Janssen Pharmaceuticals, and are employees of Janssen Pharmaceutica NV. BB and MJ have been partially supported by the Department of Health and Human Services BARDA (under the agreement HHSO100201700018C) for the submitted work, have stock and/or stock options in Johnson & Johnson, and are employees of Janssen Vaccine and Prevention B.V. No other authors have competing interests.

## Ethics committees

The COV3001 (ENSEMBLE) study was reviewed and approved by the following local ethics committees and IRBs: **Argentina:** ANMAT—Administración Nacional de Medicamentos, Alimentos y Tecnologia Médica (Capital Federal, La Plata, Ramos Mejia—Buenos Aires; Ciudad Autonoma de Buenos Aires), Comite de Etica Dr Carlos Barclay (Capital Federal, Buenos Aires; Ciudad Autonoma de Buenos Aires), Comision Conjunta de Investigacion en Salud—CCIS (La Plata, Ramos Mejia–Buenos Aires), Comite de Bioetica de Fundacion Huesped (Ciudad Autonoma de Buenos Aires), Comité de Docencia e Investigación DIM Clínica Privada (Ramos Mejia, Buenos Aires), Comité de Ética en Investigación Clínica y Maternidad Suizo Argentina (Ciudad Autonoma de Buenos Aires), Comité de Ética en Investigación de CEMIC (Ciudad Autonoma de Buenos Aires), Comite de Etica en Investigacion DIM Clinica Privada (Ramos Mejia, Buenos Aires), Comite de Etica Hospital Italiano de La Plata (La Plata, Buenos Aires), Comite de Etiica en Investigacion Hospital General de Agudos J.M. Ramos Mejia (Ciudad Autonoma de Buenos Aires), Comitéde ética del Instituto Médico Platense (CEDIMP) (La Plata, Buenos Aires), IBC Fundacion Huesped (Ciudad Autonoma de Buenos Aires), IBC Helios Salud (Ciudad Autonoma de Buenos Aires), IBC Hospital General de Agudos J.M. Ramos Mejia (Ciudad Autonoma de Buenos Aires): **Brazil:** ANVISA—Agência Nacional de Vigilância Sanitária (Salvador, Bahia; Barretos, Campinas, São Paulo, São Jose Rio Preto, Ribeirão Preto, São Caetano do Sul—São Paulo; Santa Maria, Porto Alegre—Rio Grande do Sul; Natal, Rio Grande do Norte; Para, Pará; Belo Horizonte, Minas Gerais; Rio de Janeiro, Nova Iguaçu—Rio de Janeiro; Curitiba, Paraná; Brasília, Distrito Federal; Campo Grande, Mato Grosso do Sul; Criciúma, Santa Catarina; Cuiabá, Mato Grosso), CONEP—Comissão Nacional de Ética em Pesquisa (Salvador, Bahia; São Paulo, São Paulo; Santa Maria, Rio Grande do Sul; Para, Pará;), CAPPESq —Comissão de Ética de Análise para Projetos de Pesquisa—HCFMUSP (São Paulo, São Paulo), CEP da Faculdade de Medicina de São José do Rio Preto—FAMERP (São Jose Rio Preto, São Paulo), CEP da Faculdade de Medicina do ABC/SP (São Paulo, São Paulo), CEP da Fundação Pio XII—Hospital do Câncer de Barretos/SP (Barretos, São Paulo), CEP da Liga Norteriograndense Contra o Câncer (Natal, Rio Grande do Norte), CEP da Pontificia Universidade Catolica de Campinas/PUC Campinas (Campinas, São Paulo), CEP da Real Benemérita Associaçao Portuguesa de Beneficência—Hospital São Joaquim (São Paulo, São Paulo), CEP da

Santa Casa de Misericórdia de Belo Horizonte (Belo Horizonte, Minas Gerais), CEP da Secretaria Municipal De Saúde do Rio de Janeiro—SMS/RJ (Rio de Janeiro, Rio de Janeiro), CEP da Universidade de São Caetano do Sul (CEP da Universidade de São Caetano do Sul, São Paulo), CEP da Universidade Federal de Mato Grosso do Sul—UFMS (Campo Grande, Mato Grosso do Sul), CEP da Universidade Federal de Minas Gerais (Belo Horizonte, Minas Gerais), CEP do Centro de Referência e Treinamento DST/AIDS (São Paulo, São Paulo), CEP do do INI-Ipec/Fiocruz (Rio de Janeiro, Rio de Janeiro), CEP do Grupo Hospitalar Conceição/RS (Porto Alegre, Rio Grande do Sul), CEP do Hospital das Clínicas da Faculdade de Medicina de Ribeirão Preto/USP (Ribeirão Preto, São Paulo), CEP do Hospital de Clinicas da Universidade Federal do Parana—HCUFPR/PR (Curitiba, Paraná), CEP do Hospital de Clínicas de Porto Alegre/HCPA (Porto Alegre, Rio Grande do Sul), CEP do Hospital Geral de Nova Iguaçu (Nova Iguaçu, Rio do Janeiro), CEP do Hospital Municipal São José (Criciúma, Santa Catarina), CEP do Hospital Pró-Cardíaco/RJ (Rio de Janeiro, Rio de Janeiro), CEP do Hospital Sírio Libanês (São Paulo, Sao Paulo), CEP do Hospital Universitário Júlio Muller / MT (Cuiabá, Mato Grosso), CEP do Hospital Universitário Professor Edgard Santos—UFBA (Salvador, Bahia), CEP do Instituto de Cardiologia do Distrito Federal (Brasília, Distrito Federal), CEP do Instituto de Infectologia Emílio Ribas/SP (São Paulo, Sao Paulo), CEP do Instituto de Saude e Bem Estar da Mulher—ISBEM/SP (São Paulo, Sao Paulo), CEP em Seres Humanos do HFSE—Hospital Federal dos Servidores do Estado (Rio de Janeiro, Rio de Janeiro), CONEP—Comissão Nacional de Ética em Pesquisa (Brasília, Distrito Federal, Salvador, Bahia; Belo Horizonte, Minas Gerais; Cuiabá, Mato Grosso; Campo Grande, Mato Grosso do Sul; Nova Iguaçu, Rio Janeiro—Rio Janeiro; Barretos, Campinas, Sao Jose Rio Preto, São Caetano do Sul, Sao Paulo, Ribeirão Preto—Sao Paulo; Porto Alegre, Rio Grande do Sul; Natal, Rio Grande do Norte; Curitiba, Paraná; Criciúma, Santa Catarina): **Chile:** Comité de Ética de Investigación en Seres Humanos (Santiago, Region Met), Comité Ético Científico Servicio de Salud Metropolitano Central (Santiago, Region Met), Instituto de Salud Pública de Chile (Santiago, Region Met; Talca, Temuco), Comité Ético-Científico Servicio de Salud Metropolitano Sur Oriente (Talca, Santiago), Comité de Evaluación Ética Científica Servicio de Salud Araucanía Sur Temuco (Temuco), Comité Ético Científico Servicio de Salud Metropolitano Central (Viña del Mar): **Colombia:** CEI de la Fundación Cardiovascular de Colombia (Floridablanca), Comité de Ética en Investigación Clínica de la Costa (Barranquilla), INVIMA—Instituto Nacional de Vigilancia de Medicamentos y Alimentos (Colombia) (Barranquilla), Comite de Etica en Investigacion de la E.S.E. Hospital Mental de Antioquia (Santa Marta), Comite de Etica en la Investigacion CAIMED (Bogotá), INVIMA–Instituto Nacional de Vigilancia de Medicamentos y Alimentos (Colombia) (Bogotá), Comite Corporativo de Etica en Investigacion de la Fundacion Santa Fe de Bogota (Bogotá), Comité de Ética e Investigación Biomédica de la Fundación Valle del Lili (Cali), Comite de Etica e Investigacion IPS Universitaria (Medellin), Comite de Etica en Investigacion Asustencial Cientifica de Alta Complejidad (Bogotá), Comite de Etica en Investigacion Biomedica de la Corporacion Cientifica Pediatrica de Cali (Cali), Comité de Ética en Investigación Clínica de la Costa (Barranquilla), Comite de Etica en Investigacion de la E.S.E. Hospital Mental de Antioquia (Barrio Barzal Villavicencio), Comite de Etica en Investigacion del area de la Salud de la Universidad del Norte (Barranquilla), Comite de Etica en Investigacion Medplus Centro de Recuperación Integral S.A.S (Bogotá), Comité de Ética en Investigaciones CEI-FOSCAL (Floridablanca), Comite de Etica en la Investigacion CAIMED (Bogotá), Comite de Etica para Investigacion Clinica(CEIC) de la Fundacion Centro de Investigacion Clinica CIC (Medellin), Comite de Investigaciones y Etica en Investigaciones Hospital Pablo Tobon Uribe (Medellin), INVIMA—Instituto Nacional de Vigilancia de Medicamentos y Alimentos (Colombia) (Barranquilla, Bogotá, Cali, Floridablanca, Medellin: **Mexico:** CEI del Hospital Civil de Guadalajara Fray Antonio Alcalde (Guadalajara, Jalisco), CEI Hospital La Mision (Tijuana, Baja California Norte), CI del Hospital Civil de Guadalajara Fray Antonio

Alcalde (Guadalajara, Jalisco), CI Hospital La Mision (Tijuana, Baja California Norte), Comite de Bioseguridad del Instituto Nacional de Salud Publica (Mexico, Distrito Federal; Cuernavaca, Morelos), Comite de Etica en Investigacion del Instituto Nacional de Salud Publica (Mexico, Distrito Federal; Cuernavaca, Morelos), Comité de Bioseguridad del Hospital La Misión S.A. de C.V. (Tijuana, Baja California Norte; Oaxaca, Oaxaca; Merida, Yucatán; Tijuana, Baja California Norte), Comité de Bioseguridad de la Coordinación de Investigación en Salud (IMSS) (Mexico, Estado de Mexico), Comité de Bioseguridad de Médica Rio Mayo (CLINBOR) (Mexico, Distrito Federal), Comité de Bioseguridad del Hospital Universitario "Dr. José Eleuterio González" (Monterrey, Nuevo León), COFEPRIS (Comisión Federal para la Protección contra Riesgos Sanitarios) (Cuernavaca, Morelos; Mexico, Distrito Federal; Monterrey, Nuevo León; Oaxaca, Oaxaca; Merida, Yucatán), Comite de Etica de la Fac de Med de la UANL y Hospital Universitario "Dr. Jose Eleuterio Gonzalez" (Monterrey, Nuevo León), Comite de Etica en Investigacion de la Unidad de Atencion Medica e Investigacion en Salud S.C. (Merida, Yucatán), Comite de Etica en Investigacion de Medica Rio Mayo S.C. (Mexico, Distrito Federal), Comite de Etica en Investigacion de Oaxaca Site Management Organization, S.C. (Oaxaca, Oaxaca), Comite de Etica en Investigacion del Centro Medico Nacional Siglo XXI (IMSS) (Mexico, Estado do Mexico), Comité de Investigación de la Coordinación de Investigación en Salud (IMSS) (Mexico, Estado do Mexico), Comite de Investigacion de la Unidad de Atencion Medica e Investigacion en Salud S.C. (Merida, Yucatán), Comite de Investigacion de Oaxaca Site Management Organization, S.C. (Oaxaca, Oaxaca), Comité de Investigación del Hospital Universitario José Eleuterio González (Monterrey, Nuevo León), Comite de Investigacion Medica Rio Mayo, S.C. (Mexico, Distrito Federal): **Peru:** Comite Nacional Transitorio de Etica en Invest. de los Ensayos Clinicos de la enfermedad COVID-19 (Iquitos - Maynas, Loreto; Lima, San Miguel—Lima), INS—Instituto Nacional de Salud (Peru) (Lima, San Miguel—Lima; Callao; Iquitos— Maynas, Loreto): **South Africa:** Department Agriculture, Forestry and Fisheries (DAFF) (Port Elizabeth, Mthatha—Eastern Cape; Cape Town, Worcester—Western Cape; Durban, Ladysmith, Vulindlela— KwaZulu-Natal; Johannesburg, Pretoria, Mamelodi East, Soweto, Tembisa—Gauteng; Rustenburg, Klerksdorp—North West; Bloemfontein, Free State; Middelburg, Mpumalanga; Dennilton, Limpopo), Pharma Ethics (Port Elizabeth, Eastern Cape; Durban, Ladysmith—KwaZulu-Natal; Cape Town, Western Cape; Pretoria, Mamelodi East, Johannesburg, Tembisa—Gauteng; Rustenburg, Klerksdorp—North West; Bloemfontein, Free State; Middelburg, Mpumalanga; Dennilton, Limpopo), SAHPRA—South African Health Products Regulatory Authority (Port Elizabeth, Mthatha—Eastern Cape; Cape Town, Worcester—Western Cape; Durban, Ladysmith, Vulindlela—KwaZulu-Natal; Johannesburg, Pretoria, Mamelodi East, Soweto, Tembisa—Gauteng; Rustenburg, Klerksdorp—North West; Bloemfontein, Free State; Middelburg, Mpumalanga; Dennilton, Limpopo), WIRB (Mamelodi East, Pretoria—Gauteng; Ladysmith, KwaZulu-Natal; Bloemfontein, Free State; Cape Town, Western Cape; Dennilton, Limpopo), Wits Health Consortium (Soweto, Johannesburg—Gauteng; Ladysmith, KwaZulu-Natal; Mthatha, Eastern Cape), Wits Institutional Biosafety Committee (Soweto, Pretoria, Johannesburg, Tembisa—Gauteng; Rustenburg, Klerksdorp—North West; Mthatha, Eastern Cape), University of Cape Town HREC (Cape Town, Worcester—Western Cape; University of Cape Town Institute of Infectious Disease & Molecular Medicine (Cape Town, Worcester—Western Cape), University of Cape Town Institutional Biosafety Committee (Cape Town, Worcester—Western Cape), SAMRC Human Research Ethics Committee Scientific Review (Durban, KwaZulu-Natal), Sefako Makgatho University Research Ethics Committee (SMUREC) (Pretoria, Gauteng), University of KwaZulu-Natal Institutional Biosafety Committee (Durban, KwaZulu-Natal), University of KwaZulu-Natal Ethics (Durban, Vulindlela—KwaZulu-Natal), University of Stellenbosch Ethics Committee (Cape Town, Western Cape), University of KwaZulu-Natal Institutional Biosafety Committee (Vulindlela, KwaZulu-Natal): **United States:** Advarra IBC (Detroit, MI; Chapel Hill, NC; Boston, MA; Seattle,

WA; Winston-Salem, NC; Austin, TX; Peoria, IL; Huntsville, AL; Long Beach, CA; Tucson, AZ), Biomedical Institute of New Mexico—IBC (Albuquerque, NM), Birmingham VA Medical Center—Alabama- IBC (Birmingham, AL), Clinical Biosafety Services (Hollywood, FL), Columbia University IBC (New York, NY), Copernicus Group IRB (Austin, Dallas, Houston, San Antonio—TX; Rochester, New York, Bronx, Binghamton—NY; Hillsborough, Hackensack, Newark, New Brunswick—NJ; West Palm Beach, Coral Gables, Hollywood, Miami, Orlando, Gainesville, Tampa, Hallandale Beach, Pinellas Park, The Villages, Jacksonville, Deland—FL; Fort Worth, Dallas, San Antonio—TX; Norfolk, Charlottesville—VA; Mataraie, New Orleans—LA; Nashville, Knoxville, Memphis, Bristol – TN; Cincinnati, Cleveland, Columbus, Akron—OH; Detroit, Ann Arbor, Grand Rapids—MI; Philadelphia, Pittsburgh—PA; Stanford, San Diego, San Francisco, Oakland, Long Beach, Anaheim, Sacramento, West Holly-wood—CA, Las Vegas, Reno—NV; Chicago, Peoria—IL; Omaha, NE; Mobile, Birmingham, Huntsville—AL; St Louis, Greer, Kansas City—MO; Boston, MA; Harrisburg, SD; Decatur, Atlanta, Savannah—GA; Baltimore, Rockville, Annapolis—MD; New Haven, Hartford—CT; Chapel Hill, Raleigh, Fayetteville, Charlotte, Durham, Winston-Salem—NC; Indiana-polis, Valparaiso, Evansville—IN; Seattle, WA; Aurora, CO; Lexington, Louisville—KY; Murray, West Jordan, Salt Lake City – UT; Phoenix, Tuc-son, Glendale —AZ; Spartanburg, Columbia, North Charleston, Ander-son, Charleston, Mount Pleasant—SC; Portland, Medford, Corvallis — OR; Albuquerque, Gallup—NM; Little Rock, AR; Jackson, MS; Newport News, VA, Minneapolis, MN; Lenexa, KS), WIRB (Hackensack, NJ; Dallas, TX; Baltimore, MD; Chicago, IL; Aurora, CO; Winston-Salem, NC; Min-neapolis, MN; Orlando, Miami, Gainesville – FL; Philadelphia, Pittsburgh —PA; Boston, MA; St Louis, MO; Bronx, New York, NY; New Brunswick, NJ; Phoenix, AZ; Birmingham, AL; Louisville, KY; Albuquerque, NM; New Orleans, LA; Baltimore, MD; San Francisco, CA; Tampa, FL; Aurora, CO; Columbia, SC; Decatur, GA; Reno, NV; Raleigh, NC; Little Rock, AS), Clinical Biosafety Services (Dallas, San Antonio—TX; San Diego, CA; Lexington, KY; Murray, UT; Greer, Kansas City, St Louis—MO; Rockville, MD; Las Vegas, NV; Cincinnati, Columbus, Akron—OH; Phoenix, Tucson, Glendale—AZ; North Charleston, Anderson—SC; Orlando, Pinellas Park, The Villages, Miami—FL; Birmingham, AL; Valparaiso, Evansville—IN; Lenexa, KS), Columbia University IBC (Bronx, New York), Durham VA Medical Center-IBC (Raleigh, NC), Emory University IRB (Decatur, GA), Environmental Health and Safety Office (Atlanta, GA), Institutional Bio-safety Committee (New Orleans, LA), James A. Haley Veterans Hospi-tal_IBC (Tampa, FL), Jesse Brown VA Medical Center- IBC (Chicago, IL), Mass General Brigham IBC (Boston, MA), Mount Sinai- Icahn School of Medicine IBC (New York, NY), New York Blood Center-IBC (New York, NY), OHSU IBC (Portland, OR), Partners Institutional Biosafety Commit-tee (Boston, MA), Rocky Mountain Regional VA Medical Center-IBC (Aurora, CO), Rush University Medical Center (Chicago, IL), Rush Uni-versity Medical Center-IBC (Chicago, IL), Rutgers Institutional Biosafety Committee (New Brunswick, NJ), Saint Louis University IBC (St Louis, MO), Saint Michael's Medical Center IRB (Newark, NJ), Southeast Louisiana Veterans Health Care System IBC (New Orleans, LA), St. Jude Children's Research Hospital IBC Committee (Memphis, TN), St. Jude Children's Research Hospital IRB (Memphis, TN), Stanford University Administrative Panel on Human Subjects in Medical Research (Stanford, CA), Temple University – IBC (Philadelphia, PA), The University of Chi-cago Institutional Biosafety Committee (Chicago, IL), UAMS IBC (Little Rock, AS), UIC IBC (Chicago, IL), University of Alabama at Birmingham Institutional Biosafety Committee (Birmingham, AL), University of

Arkansas IRB (Little Rock, AS), University of Kentucky Biological Safety (Lexington, KY), University of Kentucky IRB (Lexington, KY), University of Louisville IRB (Louisville, KY), University of Miami-IBC (Miami, FL), Uni-versity of Mississippi Medical Center IRB (Jackson, MI), University of Pennsylvania Institutional Biosafety Committee (Philadelphia, PA), Uni-versity of Pittsburgh IBC (Pittsburgh, Pennsylvania), University of South Florida IRB (Tampa, FL), University of Utah Institutional Biosafety Com-mittee (Salt Lake City, UT), University of Utah IRB (Salt Lake City, UT), UTHealth—IBC (Houston, TX), VA Baltimore Research & Education Foundation (BREF)- IBC (Baltimore, MD), VA Central Arkansas Veterans Healthcare System-IBC (Little Rock, AS), VA James J. Peters Department of VA Medical Center-IBC (Bronx, NY), VA Medical Center - Atlanta-IBC (Decatur, GA), VA Medical Center San Francisco- IBC (San Francisco, CA), VA North Florida/South Georgia IBC (Gainesville, FL), VA North Texas Health Care System IBC (Dallas, TX), VA San Diego Healthcare System IBC (Phoenix, AZ), VA Sierra Nevada Health Care System-IBC (Reno, NV), Vanderbilt University Instituitional Review Board (Nashville, TN), Washington University IBC (St Louis, MO), WCG IBCS (Houston, TX; Orlando, FL), Western Institutional Review Board (San Diego, CA; Detroit, MI; New Orleans, LA; New York, NY), WIRB - IBCS Services (Chicago, IL; New Orleans, LA; Oakland, CA; Minneapolis, MN; Columbus, OH; Lex-ington, KY), WJB Dorne VA Medical Center-IBC (Columbia, SC).

## Additional information

**Peer review information** *Nature Communications* thanks Sunetra Gupta, Daniel O'Connor and the other, anonymous, reviewer(s) for their con-tribution to the peer review of this work. A peer review file is available.

**Craig A. Magaret** [1], **Li Li** [1], **Allan C. deCamp** [1], **Morgane Rolland** [2,3], **Michal Juraska** [1], **Brian D. Williamson** [1,4], **James Ludwig** [1], **Cindy Molitor**[1], **David Benkeser** [5], **Alex Luedtke**[6], **Brian Simpkins** [7], **Fei Heng** [8], **Yanqing Sun** [9], **Lindsay N. Carpp** [1], **Hongjun Bai** [2,3], **Bethany L. Dearlove** [2,3], **Elena E. Giorgi**[1], **Mandy Jongeneelen**[10],

Boerries Brandenburg[10], Matthew McCallum [11], John E. Bowen[11], David Veesler [11,12], Jerald Sadoff [10], Glenda E. Gray[13,14], Sanne Roels[15], An Vandebosch[15], Daniel J. Stieh [10], Mathieu Le Gars[10], Johan Vingerhoets [15], Beatriz Grinsztejn[16], Paul A. Goepfert [17], Leonardo Paiva de Sousa [16], Mayara Secco Torres Silva[16], Martin Casapia [18], Marcelo H. Losso [19], Susan J. Little [20], Aditya Gaur[21], Linda-Gail Bekker [22], Nigel Garrett [23,24], Carla Truyers[15], Ilse Van Dromme[15], Edith Swann[25], Mary A. Marovich[25], Dean Follmann [26], Kathleen M. Neuzil [27], Lawrence Corey [1,28], Alexander L. Greninger [1,11], Pavitra Roychoudhury [1,11], Ollivier Hyrien[1] & Peter B. Gilbert [1,29,30] ✉

[1]Vaccine and Infectious Disease Division, Fred Hutchinson Cancer Center, Seattle, WA, USA. [2]US Military HIV Research Program, Walter Reed Army Institute of Research, Silver Spring, MD, USA. [3]Henry M. Jackson Foundation for the Advancement of Military Medicine, Inc, Bethesda, MD, USA. [4]Biostatistics Division, Kaiser Permanente Washington Health Research Institute, Seattle, WA, USA. [5]Departments of Biostatistics and Bioinformatics, Rollins School of Public Health, Emory University, Atlanta, GA, USA. [6]Department of Statistics, University of Washington, Seattle, WA, USA. [7]Department of Computer Science, Pitzer College, Claremont, CA, USA. [8]University of North Florida, Jacksonville, FL, USA. [9]University of North Carolina at Charlotte, Charlotte, NC, USA. [10]Johnson & Johnson Innovative Medicine, Janssen Vaccines & Prevention B.V, Leiden, The Netherlands. [11]Department of Biochemistry, University of Washington, Seattle, WA, USA. [12]Howard Hughes Medical Institute, University of Washington, Seattle, WA, USA. [13]Perinatal HIV Research Unit, Faculty of Health Sciences, University of the Witwatersrand, Johannesburg, South Africa. [14]South African Medical Research Council, Cape Town, South Africa. [15]Janssen R&D, a division of Janssen Pharmaceutica NV, Beerse, Belgium. [16]Evandro Chagas National Institute of Infectious Diseases-Fundação Oswaldo Cruz, Rio de Janeiro, RJ, Brazil. [17]Division of Infectious Diseases, Department of Medicine, University of Alabama at Birmingham, Birmingham, AL, USA. [18]Facultad de Medicina Humana, Universidad Nacional de la Amazonia Peru, Iquitos, Peru. [19]Hospital General de Agudos José María Ramos Mejia, Buenos Aires, Argentina. [20]Division of Infectious Diseases, University of California San Diego, La Jolla, CA, USA. [21]Department of Infectious Diseases, St. Jude Children's Research Hospital, Memphis, TN, USA. [22]The Desmond Tutu HIV Centre, University of Cape Town, Observatory, Cape Town, South Africa. [23]Centre for the AIDS Programme of Research in South Africa, University of KwaZulu-Natal, Durban, South Africa. [24]Discipline of Public Health Medicine, School of Nursing and Public Health, University of KwaZulu-Natal, Durban, South Africa. [25]Vaccine Research Program, Division of AIDS, National Institute of Allergy and Infectious Diseases, National Institutes of Health, Bethesda, MD, USA. [26]Biostatistics Research Branch, National Institute of Allergy and Infectious Diseases, National Institutes of Health, Bethesda, MD, USA. [27]Center for Vaccine Development and Global Health, University of Maryland School of Medicine, Baltimore, MD, USA. [28]Department of Laboratory Medicine and Pathology, University of Washington, Seattle, WA, USA. [29]Public Health Sciences Division, Fred Hutchinson Cancer Center, Seattle, WA, USA. [30]Department of Biostatistics, University of Washington School of Public Health, Seattle, WA, USA. ✉e-mail: pgilbert@fredhutch.org

