## [Peer Review File · Nature Communications]

Quantifying how single dose Ad26.COV2.S vaccine efficacy depends on Spike sequence featuresReviewers' Comments:

Reviewer #1:

Remarks to the Author:

This is a very useful study which asks many important questions concerning the role of vaccine induced immunity in preventing infection and severe attendant outcomes; these results also have implications for the role of naturally acquired immunity in combatting coronavirus and for identifying the critical targets of B cell responses.

I am concerned however that Latin America has been amalgamated into a single category when comparing with the USA and RSA and I was surprised not to see a breakdown at a country level of this data. My count of the available samples reads as: Brazil – 245, Argentina – 123, Peru- 107, Colombia – 239, Chile – 12, RSA – 24, USA – 313. This indicates that there is no particular reason not to disaggregate the Latin American samples into countries.

The results generally endorse the hypothesis that vaccination will durably protect against severe outcomes while unlikely to protect against reinfection, as might be expected of any member of the coronavirus family. It is difficult to deconvolute the contribution of antigenic change to either of these processes from the natural decay of infection-blocking immunity as compared to the near permanence of immunity against severe outcomes. It is important that the discussion should contain this point.

I was particularly impressed by the attempts to assess the relative importance of variable residues, and I would suggest that figs S31 and S32 be moved into the main text to illustrate these points.

I also have some issues concerning Fig 1. As mentioned already, I think the Latin America data needs to be broken down by country. The tree is impossible to read and may be best to move into Supp Mat. Figure C would be better rotated by 90 degrees to avoid confusing the x axis with time.

In Fig 3 (and attendant analyses), perhaps it would be better to leave out the lambda and zeta outliers resulting from large deletions – or at least show the results with and without them included. Have you tried using AlphaFold to see what effect they have? From Fig S32 it would appear that they do not contain any antigenic sites.

Reviewer #2:

Remarks to the Author:

This manuscript aims to identify the SARS-CoV-2 sequence features that define breakthrough disease in vaccinated individuals. The methods described are interesting and the sample size is large and from a well-controlled trial (RCT). The major issue I have with this paper is I find it hard to relate to these findings to the currently circulating variants (Omicron subvariants) and in the context of hybrid immunity? Also, the selection of “moderate-severe” cases is also puzzling, as presumably there are more data from all symptomatic cases than just this subset? As a result, I am unclear what the conclusion of the study is? The last sentence of the abstract is “These results help map antigenic specificity of in vivo vaccine protection” which is also not a conclusion...

Minor points

Abstract

- “It is of interest to pinpoint SARS-CoV-2 sequence features defining vaccine resistance”, rather than resistance isn’t this escape or breakthrough infection?
- What is the definition of “moderate to severe-critical” not just referencing previous paper?
 - o Also need a table of disease /demographics
- I don’t see the point in —Fig. 2B shows VE against the primary COVID-19 endpoint caused by the

groupings of all other lineages excluding each individual lineage ("not-lineage").

- Would move some of the stats into the main text, e.g., the q-value/p-value significance level used — unusual to have to review the supplementary for this ...
- It's unclear what physicochemical attributes were incorporated in the "weighted hamming distance"?
- "Tables S10 and S11 show inferences about differences in mean escape scores of vaccine vs. placebo sequences" — these sentences do not make sense.
- Typo bottom of page 14 — "417 twchich is a characteristic..."

Reviewer #3:

Remarks to the Author:

This manuscript provides insightful statistical analyses to identify the role of Spike sequence features in the vaccine efficacy (VE) of an Adenovirus-vectored SARS-CoV-2 vaccine in a phase 3 clinical trial. It features carefully detailed and rigorous statistical analyses, and the findings are significant to the field.

While the methodology employed by the authors is sound, the interpretation and analyses are hampered the following major issue. The authors correctly identify and note that in the Latin American arm of the trial, the significant Spike sequence features are mainly driven by the reduced VE against Lambda variants. Consequently, any Spike mutation enriched in Lambda is identified as a significant sieve effect site. However, not all such mutations would be "drivers" - for example resistance mutations to antibody binding/neutralizing responses - and quite possibly several of these will likely be "bystanders". The authors do not attempt to try and disentangle such mutations, which is a significant limitation of this study to this reviewer. For example, the authors could remove Lambda breakthrough infections and explore which mutations are significant in the other variants, and if they find any, they could explore how such mutations could or could not explain the reduced VE against Lambda. Another alternative is to use the approach of Korber et al. (<https://pubmed.ncbi.nlm.nih.gov/17363674/>, <https://pubmed.ncbi.nlm.nih.gov/30629920/>) who developed a phylogenetically corrected way of identifying sequence associations that overcomes the issue of "bystander" mutations. Addressing this limitation will significantly enrich the manuscript as it can lead to pin-pointing potentially driver mutations underlying VE reduction. Even if such analyses do not yield any significant resistance mutations, or mutations that cannot fully explain the low VE against Lambda, they would still improve the manuscript as they will show more conclusively that in the set of novel Lambda mutations are mutation(s) that drive the reduced VE, but whose identity remains cannot be determined using such analyses. Thus, I strongly urge the authors to consider addressing this issue using any method of their choice.

The following issues will help improve readability of this manuscript:

- 1) Figure S4 is quite telling and I would urge the authors to include it as a panel in a main figure.
- 2) Lines 150-3: It is not clear if the results for the 38 residues are shown in any supplementary item. If not, please consider displaying these results.
- 3) Lines 189-221: Current phrasing is unclear if the DMS studies were done in this work, or previously. The supplementary information does not describe how the different DMS maps were obtained. I am assuming that these are from different NABs or sera, but it would help to add a column to Table S1 indicating which NABs or sera gave rise to each DMS cluster. It will help readers to be able to connect the DMS/PDB clusters to known NAB classification schemes such as Barnes et al. Class 1-4 (<https://pubmed.ncbi.nlm.nih.gov/33045718/>) or Hastie et al. (<https://pubmed.ncbi.nlm.nih.gov/34554826/>) and/or giving examples of antibodies that typify interesting clusters. Because such antibody classes or monoclonals will be more informative to readers, I would also recommend using these clusters or specific monoclonal antibody names in the

nomenclature of the DMS cluster, for example "DMS1 (Barnes et al. Class XXX)" or "DMS2 (monoclonal XXX derived)", etc.

5) Lines 225-9: The motivation and description of these NTD features is buried in the supplement and SAP. Please consider briefly mentioning these in the main text, and using descriptive names instead of "NTD1"- "NTD7" (e.g. "Indel 138-158/242-264" for NTD1, or "Mutation 12-13" for NTD2) such that readers can identify the Spike sites.

6) Lines 245-248: Please consider briefly summarizing results from the variable importance analyses in the main text.

7) Discussion: It is clear that Lambda drives the sequence features impacting VE -- the limitation that there could be bystander mutations in the set of mutations identified is not discussed. The interesting finding that perhaps Lambda neutralization resistance might not capture the primary endpoint VE for Lambda is not discussed.

REVIEWER COMMENTS

Reviewer #1 (Remarks to the Author):

This is a very useful study which asks many important questions concerning the role of vaccine induced immunity in preventing infection and severe attendant outcomes; these results also have implications for the role of naturally acquired immunity in combatting coronavirus and for identifying the critical targets of B cell responses.

Response: Thank you for the positive feedback.

I am concerned however that Latin America has been amalgamated into a single category when comparing with the USA and RSA and I was surprised not to see a breakdown at a country level of this data. My count of the available samples reads as: Brazil – 245, Argentina – 123, Peru- 107, Colombia – 239, Chile – 12, RSA – 24, USA – 313. This indicates that there is no particular reason not to disaggregate the Latin American samples into countries.

Response: The aggregation of the Latin American samples into a single category for the sieve analyses was motivated by the increase in statistical power that is achieved by adding endpoints. Moreover, overall differences were seen in the strains that were circulating across Latin America vs. RSA vs. the USA (“Five main variants circulated in Latin America (Reference, Zeta, Gamma, Lambda, Mu), while the South African sequences were 76% Beta and 17% Delta, and the US sequences were 85% Reference (Fig. 1a).”).

However, we take the reviewer’s point that there is value in also presenting country-specific information for Latin America. We have made the following revisions to the manuscript: First, we reworked Supplementary Table 6 to provide a country-specific breakdown of the lineages that caused the primary endpoints.

Second, we performed four different Latin America country-specific sieve analyses, and integrated these results into the supplementary material of the manuscript. These analyses include assessment VE against the primary COVID-19 endpoint caused by specific lineages (Fig. R1), by all other lineages combined (Fig. R2), differential VE estimates against COVID-19 by pairs of lineage or across a lineage vs. all other lineages (Fig. R3), and VE against the primary COVID-19 endpoint with genotypes defined by individual spike AA position residues matching vs. mismatching the vaccine strain (Fig. R4). The same screening rule as the Latin America analysis was applied, i.e., lineages with at least 20 COVID-19 endpoints were included, and amino acid positions with at least 20 vaccine-mismatched COVID-19 endpoints were included. Note that Chile and Peru were combined in this analysis, due to the fact that there were too few endpoints ($n = 12$) in Chile for its own analysis, as well as the fact that, among Chile’s geographic neighbors, Peru had slightly fewer cases than Argentina ($n = 121$ vs. 131).

The country-specific sieve analysis affords fewer pairwise comparisons of VE against lineages and fewer AA positions in site-scanning. For example, the differential VE of Lambda vs. Zeta, Lambda vs. Mu, Zeta vs. Mu can no longer be estimated (Fig. R3). Among the VEs that were estimated, the confidence intervals are much wider, compared to those based on Latin America analysis, due to a reduction of sample sizes. The trends in the estimated VE against primary COVID-19 endpoint caused

by lineages in each country agree with those based on Latin America analysis in general. For example, VE is higher against Reference than against Lambda and against not-Reference lineages (Fig. R3). VE is also higher against not-Lambda vs. Lambda, against Zeta vs. Lambda, against Reference vs. Gamma, Reference vs. Mu, and Zeta vs. Gamma. We added Figs R1-R3 to the revision as Supplementary Figs. 5-7, except without the Latin America overall panel for comparison as that is shown in the main text.

A) Latin America

B) Argentina

C) Brazil

D) Chile and Peru

E) Colombia

Figure R1. For (A) Latin America, (B) Argentina, (C) Brazil, (D) Chile and Peru, and (E) Colombia, vaccine efficacy (VE) estimates against the primary COVID-19 endpoint caused by SARS-CoV-2 lineages (lineage “X”). To avoid posthoc revision of the SAP, p values were not calculated for the country-specific analyses. Panel A is identical to Fig. 2A in the manuscript and is shown for comparison.

A) Latin America

B) Argentina

C) Brazil

D) Chile and Peru

E) Colombia

Figure R2. For (A) Latin America, (B) Argentina, (C) Brazil, (D) Chile and Peru, and (E) Colombia, vaccine efficacy (VE) estimates against the primary COVID-19 endpoint caused by all other lineages combined (“Not X”). To avoid posthoc revision of the SAP, p values were not calculated for the country-specific analyses. Panel A is identical to Fig. 2B in the manuscript and is shown for comparison.

A) Latin America

Comparison	Differential VE* (95% CI)	Two-sided Differential VE		
		P-value ^c	FWER P-value	Q-value
Reference vs. Not Reference ^{¶§†}	1.58 (1.16, 2.17)	0.0041	0.05	0.015
Gamma vs. Not Gamma	0.75 (0.54, 1.06)	0.10	0.60	0.15
Lambda vs. Not Lambda ^{¶§†}	0.49 (0.31, 0.76)	0.0015	0.019	0.0073
Mu vs. Not Mu	0.77 (0.50, 1.18)	0.23	0.73	0.27
Zeta vs. Not Zeta	1.45 (0.96, 2.20)	0.08	0.56	0.13
Reference vs. Gamma ^{¶§}	1.68 (1.14, 2.49)	0.0087	0.096	0.026
Reference vs. Lambda ^{¶§†}	2.52 (1.55, 4.11)	<0.001	0.0031	0.0031
Reference vs. Mu ^{¶§}	1.71 (1.06, 2.75)	0.028	0.28	0.066
Reference vs. Zeta	0.99 (0.62, 1.58)	0.97	1	0.97
Gamma vs. Lambda	1.50 (0.90, 2.50)	0.12	0.61	0.17
Gamma vs. Mu	1.01 (0.62, 1.67)	0.96	1	0.97
Gamma vs. Zeta ^{¶§}	0.59 (0.36, 0.95)	0.031	0.28	0.066
Lambda vs. Mu	0.68 (0.38, 1.20)	0.18	0.73	0.23
Lambda vs. Zeta ^{¶§†}	0.39 (0.22, 0.69)	0.0011	0.016	0.0073
Mu vs. Zeta	0.58 (0.33, 1.01)	0.054	0.43	0.10

*Differential VE (DVE) for genotype 1 vs. genotype 2, with VE(genotype 1) >= VE(genotype 2), is calculated as DVE = [1 - VE(genotype 2)]/[1 - VE(genotype 1)], with interpretation that vaccine protection is DVE-fold better against genotype 1 than against genotype 2.

[¶] Unadjusted P-value for differential VE is ≤ 0.05.

[†] FWER-adjusted P-value for differential VE is ≤ 0.05.

[§] FDR-adjusted P-value (Q-value) for differential VE is ≤ 0.2 and unadjusted P-value for differential VE is ≤ 0.05.

B) Argentina

Comparison	Differential VE (95% CI)
Reference vs. Gamma	1.43 (0.61, 3.36)
Reference vs. Lambda	1.51 (0.52, 4.35)
Gamma vs. Lambda	1.06 (0.35, 3.16)
Reference vs. Not Reference	1.59 (0.76, 3.33)
Gamma vs. Not Gamma	0.88 (0.41, 1.92)
Lambda vs. Not Lambda	0.85 (0.32, 2.25)

C) Brazil

Comparison	Differential VE (95% CI)
Reference vs. Gamma	2.51 (1.19, 5.29)
Reference vs. Zeta	1.01 (0.47, 2.14)
Gamma vs. Zeta	0.40 (0.23, 0.70)
Reference vs. Not Reference	1.51 (0.75, 3.01)
Gamma vs. Not Gamma	0.39 (0.23, 0.65)
Zeta vs. Not Zeta	1.80 (1.08, 3.00)

D) Chile and Peru

Comparison	Differential VE (95% CI)
Reference vs. Gamma	0.95 (0.30, 3.01)
Reference vs. Lambda	2.79 (1.19, 6.56)
Gamma vs. Lambda	2.95 (1.06, 8.17)
Reference vs. Not Reference	1.98 (0.87, 4.47)
Gamma vs. Not Gamma	1.95 (0.73, 5.20)
Lambda vs. Not Lambda	0.34 (0.16, 0.69)

E) Colombia

Comparison	Differential VE (95% CI)
Reference vs. Gamma	1.63 (0.69, 3.83)
Reference vs. Mu	1.76 (1.01, 3.05)
Gamma vs. Mu	1.08 (0.44, 2.62)
Reference vs. Not Reference	1.68 (1.01, 2.80)
Gamma vs. Not Gamma	0.79 (0.35, 1.80)
Mu vs. Not Mu	0.62 (0.37, 1.06)

Figure R3. For (A) Latin America, (B) Argentina, (C) Brazil, (D) Chile and Peru, and (E) Colombia, differential vaccine efficacy (VE) estimates against the primary COVID-19 endpoint across pairs of lineages or across a lineage (“X”) vs. all other lineages (“Not X”). To avoid posthoc revision of the SAP, p values were not calculated for the country-specific analyses. Panel A is identical to Fig. 2C in the manuscript and is shown for comparison.

For the 4 positions (75, 76, 253, 490) with FWER p ≤ 0.05 based on Latin America analysis, the trends are also similar in the country-specific analysis, i.e., VE is higher against an endpoint with a vaccine matching genotype (Fig. R4; added to the revision as Supplementary Figs. 9-12).

The results generally endorse the hypothesis that vaccination will durably protect against severe outcomes while unlikely to protect against reinfection, as might be expected of any member of the coronavirus family. It is difficult to deconvolute the contribution of antigenic change to either of these processes from the natural decay of infection-blocking immunity as compared to the near permanence of immunity against severe outcomes. It is important that the discussion should contain this point.

Response: Thank you for this comment. We have revised the Discussion to include this point and have additionally cited a relevant Review by Milne et al. (Lancet Respir Med 2021):

“Thus, the results generally endorse the hypothesis that a single dose of Ad26.COVS vaccine will durably protect against severe outcomes. However, it is unlikely to protect against reinfection, as might be expected of any member of the coronavirus family.³⁹ It is difficult to deconvolute the effects of antigenic change on protection from severe outcomes and protection from reinfection from the natural decay of infection-blocking immunity as compared to the near permanence of immunity against severe outcomes.” (p. 21)

I was particularly impressed by the attempts to assess the relative importance of variable residues, and I would suggest that figs S31 and S32 be moved into the main text to illustrate these points.

Response: We have done this, where they are now Fig. 5g-h and Fig. 6b-e.

I also have some issues concerning Fig 1. As mentioned already, I think the Latin America data needs to be broken down by country. The tree is impossible to read and may be best to move into Supp Mat. Figure C would be better rotated by 90 degrees to avoid confusing the x axis with time.

Response: We agree that the phylogenetic tree is difficult to read due to the density of information. To accommodate the legibility of these large trees, we followed your suggestion to move the Latin America tree (previous Fig. 1b) to the supplement, and we have additionally refactored this tree to indicate country and variant information by color. This figure is now Supplementary Fig. 1. We also generated trees for the other regions (South Africa, United States), each of which is also its own supplementary figure (Supplementary Figs. 2, 3 in the revision).

With the phylogenetic tree removed from Fig. 1, we had space to lay out panels (a) and (b) horizontally. We believe this avoids visual comparison across the two x-axes and helps avoid confusing the x axis in the new panel (b) with time.

In Fig 3 (and attendant analyses), perhaps it would be better to leave out the lambda and zeta outliers resulting from large deletions – or at least show the results with and without them included. Have you tried using AlphaFold to see what effect they have? From Fig S32 it would appear that they do not contain any antigenic sites.

Response: This is a good suggestion. We performed a sensitivity analysis of VE by physicochemical Hamming distances without the Lambda and Zeta outliers, with the results shown below in Figure R5 (also in the revision as Supplementary Fig. 16). The results agree with the Latin America analysis that vaccine efficacy against COVID-19 decreases with increasing protein distance to the vaccine-strain in

Spike, receptor-binding domain (RBD), N-terminal domain (NTD), and S1. After removing the outliers, Lambda virus had 17.2-18.2 mismatches and Zeta had 8.1- 10.6 mismatches. The ordering of lineages by protein distance and the ordering of the VE estimates by lineage category remain the same as that of the Latin America analysis. We have added a sentence summarizing this result to the main text:

“A sensitivity analysis of VE by physicochemical-weighted Hamming distance performed after removing the Lambda and Zeta outliers (defined by distance > 20) in Fig. 3 yielded similar results (Supplementary Fig. 16).”

Figure R5. For the Latin America cohort, sensitivity analysis of vaccine efficacy (VE) against the primary COVID-19 endpoint by physicochemical-weighted Hamming distances in (A) Spike, (B) the RBD domain, (3) the NTD domain, or (4) the S1 region of the disease-causing SARS-CoV-2 isolate to that of the vaccine-insert sequence. The analysis was performed after deleting observations with Spike physicochemical-weighted Hamming distance greater than 20 (11 observations corresponding to Zeta and Lambda). The top plot in each panel shows the distributions of distances by treatment arm, color-coded by lineage. The bottom plot in each panel shows the estimated VE by SARS-CoV-2 sequence distance. The dotted lines are pointwise 95% confidence intervals. The dots are overall VE estimates for the given lineage placed at the lineage-specific median distance of placebo arm endpoints, with vertical bars indicating their pointwise 95% confidence intervals.

Reviewer #2 (Remarks to the Author):

This manuscript aims to identify the SARS-CoV-2 sequence features that define breakthrough disease in vaccinated individuals. The methods described are interesting and the sample size is large and from a well-controlled trial (RCT).

The major issue I have with this paper is I find it hard to relate to these findings to the currently circulating variants (Omicron subvariants) and in the context of hybrid immunity?

Thank you for the question. The original Discussion had a paragraph describing the association between our sieve findings and the then-current state of the pandemic (“The Ad26.COVS vaccine sieve effects observed here, based on data collected prior to July 10, 2021, revealed broader vaccine adaptation features as several sieve signature sites showed mutations in subsequent variant waves. ...”), but a lot has changed since then. This text has been expanded into three paragraphs and updated to reflect the current epidemic. The updated text can be found on pp. 21-22 of the revision.

The question of hybrid immunity is extremely relevant, given that as the pandemic progressed, high global population rates of SARS-CoV-2 seropositivity were eventually reached.¹⁻⁵ Our analysis was conducted in baseline SARS-CoV-2 seronegative individuals to mirror the population for primary analyses of vaccine efficacy (Sadoff et al. 2021, 2022 NEJM).^{6,7} Moreover, at the time during which the study was conducted, baseline seropositivity rates were much lower than they are at present day, where the vast majority of COVID-19 cases with SARS-CoV-2 sequence data in the data set being baseline-seronegative participants. There are not enough baseline-seropositive participants with sequence data to be able to investigate how VE varied by virus features in baseline-seropositive participants; this is the reason the manuscript restricts to baseline-seronegative participants.

Moyo-Gwete et al. have shown that infection prior to Ad26.COVS vaccination (i.e. hybrid immunity) results in superior responses compared to vaccination alone.⁸ Specifically, a single Ad26.COVS dose administered to healthcare workers in South Africa elicited durable (up to 6 months post-vaccination) binding antibodies, Fc functionality (antibody-dependent cellular cytotoxicity), neutralizing antibodies, and cellular responses (CD4+ and CD8+) to the Ancestral strain. Moreover, most of these antibody responses to the Ancestral strain were significantly higher in individuals with hybrid immunity compared to individuals without prior infection, and cross-reactivity against the Ancestral, Beta, Delta, and Omicron variants was also greater in the hybrid immunity cohort. It remains an open question to be answered in future investigations whether and how these increased responses and greater cross-reactivity would impact the sieve effects observed in the present study.

Also, the selection of “moderate-severe” cases is also puzzling, as presumably there are more data from all symptomatic cases than just this subset?

For the primary efficacy analysis (Sadoff et al., NEJM 2022), the primary endpoints were moderate to severe-critical COVID-19 with onset at least 14 days after administration and with onset at least 28 days after administration.

In the primary efficacy analysis, only 1 case was mild out of 117 symptomatic COVID-19 events in the vaccine group and only 3 of 351 in the placebo group. Thus, the vast majority of symptomatic COVID-19 events were moderate to severe-critical.

As a result, I am unclear what the conclusion of the study is? The last sentence of the abstract is “These results help map antigenic specificity of in vivo vaccine protection” which is also not a conclusion...

Response: We have removed this sentence from the Abstract and expanded upon the Discussion section, to emphasize the primary findings of Lambda being an escape variant and that the Ad26.COVS vaccine protects well against severe/critical disease.

Minor points

Abstract

- “It is of interest to pinpoint SARS-CoV-2 sequence features defining vaccine resistance”, rather than resistance isn’t this escape or breakthrough infection?

Response: We have removed this sentence from the Abstract.

- What is the definition of “moderate to severe–critical” not just referencing previous paper?

Response: We have added the definitions to the Methods section:

“Moderate COVID-19 was defined by a positive RT-PCR test for SARS-CoV-2 as well as two or more of the following symptoms (new or worsening): fever or chills, cough, heart rate ≥ 90 beats/minute, muscle or body pain, headache, new loss of taste or smell, sore throat, red or bruised-looking feet or toes, nausea, vomiting, or diarrhea; or one or more of the following signs or symptoms: shortness of breath, respiratory rate > 20 breaths/minute, clinical or radiologic evidence of pneumonia, deep vein thrombosis, or abnormal oxygen saturation (but above 93%).^{10,41}

Severe–critical COVID-19 was defined by a positive RT-PCR test for SARS-CoV-2 with one of the following features: respiratory failure; evidence of shock (systolic blood pressure < 90 mm Hg, diastolic blood pressure < 60 mm Hg, or requiring vasopressors); respiratory rate > 30 breaths/minute; heart rate ≥ 125 beats/minute; oxygen saturation of 93% or less (ambient air at sea level), or a ratio of the partial pressure of oxygen to the fraction of inspired oxygen < 300 mm Hg; intensive care unit admission; significant acute renal, hepatic, or neurologic dysfunction, or death.^{10,41}”

o Also need a table of disease /demographics

Response: We have added Table 2, which reports participant demographics in the Latin America cohort, as a main item to the revised manuscript. We have also added Supplementary Tables 7-9, which provide equivalent information for the United States, South Africa, and geographic regions-pooled cohorts, respectively.

- I don't see the point in —Fig. 2B shows VE against the primary COVID-19 endpoint caused by the groupings of all other lineages excluding each individual lineage (“not-lineage”).

Response: With some of our analyses, we compare two lineages directly (e.g., Reference vs. Zeta) when we want to tease out the differences between them, but in other analyses we want to compare the sequences from a specific lineage with all other sequences in the study, to see if there is a difference that is specific to that lineage only. We accomplish this with the “lineage” vs. “not-lineage” analyses: they are comparing the “lineage” sequences with everything else.

- Would move some of the stats into the main text, e.g., the q-value/p-value significance level used — unusual to have to review the supplementary for this ...

Response: We agree that much of the Supplementary Materials and Methods could be moved to the main text, and that it would be beneficial to the reader to do so. We have made the following revisions:

-Expanded the following existing sections by moving a portion of the supplementary text to the main text Methods: “SARS-CoV-2 sequencing and sequence data”, “Sieve analysis”.

-Moved the following sections (completely) from Supplementary Materials and Methods to the main text Methods: “Specification of Spike AA sequence features for sieve analysis”, “Handling of missing sequences”, “Quantification of viral diversity”, “Structural modeling”, “AA sequence sieve analysis methods: Prospective VE sieve analysis”, “Multiple hypothesis testing adjustment for AA sequence sieve analysis”, “Neutralization hypothesis-driven sieve analysis”, and “Covariability analysis”.

-Moved portions of existing Supplementary Materials and Methods sections to the main text Methods: “Deep mutational scanning (DMS) antibody escape scores”, “Protein Data Bank (PDB) antibody escape scores”.

Note that the info on q/value/p-value significance level used is in the “Multiple hypothesis testing adjustment for AA sequence sieve analysis” section, and is thus now in the main text as requested.

- It's unclear what physicochemical attributes were incorporated in the “weighted hamming distance”?

Response: Thank you for bringing this to our attention. We added a new section in the Supplementary Materials and Methods, “Calculation of Physicochemically Weighted Hamming Distances”, which explains this process.

- “Tables S10 and S11 show inferences about differences in mean escape scores of vaccine vs. placebo sequences” — these sentences do not make sense.

Response: We have revised this text to the following:

DMS escape scores: “Supplementary Tables 14 and 15 show the mean DMS escape scores in the vaccine arm and in the placebo arm of the disease-causing SARS-CoV-2 isolates, as well as the difference in mean DMS escape score between the two arms. To accommodate for missing sequences, in the analysis whose results are shown in Supplementary Table 14, doubly robust targeted minimum loss-based estimation was used; in Supplementary Table 15, inverse probability weighting was used. Results were generally similar using the two different statistical methods, with greater mean DMS escape scores in the vaccine arm than the placebo arm for all of the clusters and the lower limit of the 95% CI usually greater than zero. The greatest difference in mean DMS escape score (vaccine – placebo) was seen for DMS5 [0.051 (0.0032, 0.098) in Supplementary Table 14, 0.13 (0.073, 0.19) in Supplementary Table 15]. Geographic region-specific results are also shown in Supplementary Tables 14 and 15.”

PDB escape scores: “Supplementary Tables 17 and 18 show the mean PDB escape scores in the vaccine arm and in the placebo arm of the disease-causing SARS-CoV-2 isolates, as well as the difference in mean PDB escape score between the two arms. Analyses were performed the same as for Supplementary Tables 14 and 15. For each cluster, the mean PDB escape score was generally higher in the vaccine arm than in the placebo arm. The greatest difference in mean PDB escape score (vaccine – placebo) was seen for PDB13 [0.27 (0.043, 0.5) in Supplementary Table 17 and 0.47 (0.24, 0.7) in Supplementary Table 18]. Geographic region-specific results are also shown in Supplementary Tables 17 and 18.”

- Typo bottom of page 14 — “417 twchich is a characteristic...”

Response: Fixed, thank you.

Reviewer #3 (Remarks to the Author):

This manuscript provides insightful statistical analyses to identify the role of Spike sequence features in the vaccine efficacy (VE) of an Adenovirus-vectored SARS-CoV-2 vaccine in a phase 3 clinical trial. It features carefully detailed and rigorous statistical analyses, and the findings are significant to the field.

Response: Thank you for the positive comments.

While the methodology employed by the authors is sound, the interpretation and analyses are hampered the following major issue. The authors correctly identify and note that in the Latin American arm of the trial, the significant Spike sequence features are mainly driven by the reduced VE against Lambda variants. Consequently, any Spike mutation enriched in Lambda is identified as a significant sieve effect site. However, not all such mutations would be "drivers" - for example resistance mutations to antibody binding/neutralizing responses - and quite possibly several of these will likely be "bystanders". The authors do not attempt to try and disentangle such mutations, which is a significant

limitation of this study to this reviewer. For example, the authors could remove Lambda breakthrough infections and explore which mutations are significant in the other variants, and if they find any, they could explore how such mutations could or could not explain the reduced VE against Lambda. Another alternative is to use the approach of Korber et al. (<https://pubmed.ncbi.nlm.nih.gov/17363674/>, <https://pubmed.ncbi.nlm.nih.gov/30629920/>) who developed a phylogenetically corrected way of identifying sequence associations that overcomes the issue of "bystander" mutations. Addressing this limitation will significantly enrich the manuscript as it can lead to pinpointing potentially driver mutations underlying VE reduction. Even if such analyses do not yield any significant resistance mutations, or mutations that cannot fully explain the low VE against Lambda, they would still improve the manuscript as they will show more conclusively that in the set of novel Lambda mutations are mutation(s) that drive the reduced VE, but whose identity remains cannot be determined using such analyses. Thus, I strongly urge the authors to consider addressing this issue using any method of their choice.

Response: We took the reviewer's helpful suggestion to remove Lambda COVID-19 endpoints in the amino acid site-scanning analyses for the Latin America data. The results, displayed below in Tables R1, R2, and R3, show that the number of screened-in AA sequence features decreased from 37 (as shown in Supplementary Fig. 15) to 24. The 13 AA positions that were covariable with Lambda and were significant with FDR q-value ≤ 0.2 in the original results that included all lineages (Fig. 2d) were no longer screened in.

Supplementary Fig 15 shows the results of the 37 screened-in sites in the all-lineage analysis, and Table R1 below shows the results of the 24 screened-in sites in the Lambda-removed analysis. Of the 16 AA positions with sieve effect evidence (Fig. 2d), 3 qualified for the sensitivity analysis based on sufficient residue variability (sites 414, 501, 778), and the results for these three sites are shown in Table R2. Without the Lambda lineage, vaccine efficacy for vaccine-matched residue COVID-19 and vaccine-mismatched residue COVID-19, respectively, at these three sites were 53.2 (95% CI 44.9, 60.2%) and 87.5% (64.5, 95.6%) for position 414; 65.0 (95% CI 56.2, 72.0%) and 40.5% (24.8, 52.9%) for position 501; 53.7 (95% CI 45.5, 60.6%) and 95.1% (63.8, 99.3%) for position 778 (Tables R1, R2). P-values and q-values were not calculated for these results, as the analysis was not pre-specified. These results are very similar to the results that included all lineages as reported in Fig. 2d of the manuscript (comparison shown in Table R2B). We summarize the results of these exploratory analyses in a new subsection to Results titled "Exploratory analyses of Lambda's escape and driver mutations" (pp. 15-17 of the revision), and additionally included the new Supplementary Fig. 50 in the revision.

Table R1. Estimation and inference in the per-protocol cohort about VE against the primary (moderate to severe-critical) COVID-19 endpoint with match or mismatch to the vaccine insert at all screened-in positions in Latin America. This analysis restricts cases to lineages that are not Lambda.

Spike Amino Acid Position	No. of Cases (V vs. P) (Incidence per 100 PYRs)	VE (%) (95% CI)	
18			
Match	144 (5.1) vs. 353 (12.8)	61.2 (52.8, 68.0)	
Mismatch	76 (2.7) vs. 115 (4.2)	38.0 (17.1, 53.6)	
20			
Match	147 (5.2) vs. 359 (13)	61.0 (52.7, 67.8)	
Mismatch	73 (2.6) vs. 109 (4)	37.3 (15.7, 53.4)	
26			
Match	147 (5.2) vs. 357 (13)	60.8 (52.5, 67.6)	
Mismatch	73 (2.6) vs. 111 (4)	38.4 (17.2, 54.2)	
69			
Match	214 (7.5) vs. 451 (16.4)	54.9 (47.0, 61.7)	
Mismatch	6 (0.2) vs. 17 (0.6)	68.0 (18.9, 87.4)	
70			
Match	214 (7.5) vs. 451 (16.4)	54.9 (47.0, 61.7)	
Mismatch	6 (0.2) vs. 17 (0.6)	68.0 (18.9, 87.4)	
95			
Match	177 (6.2) vs. 401 (14.6)	57.5 (49.3, 64.4)	
Mismatch	43 (1.5) vs. 67 (2.4)	43.8 (17.6, 61.7)	
138			
Match	145 (5.1) vs. 353 (12.8)	60.9 (52.5, 67.8)	
Mismatch	75 (2.6) vs. 115 (4.2)	38.8 (18.2, 54.3)	
144			
Match	170 (6) vs. 385 (14)	57.4 (49.0, 64.4)	
Mismatch	50 (1.8) vs. 83 (3)	46.7 (24.3, 62.5)	
145			
Match	183 (6.4) vs. 414 (15)	57.5 (49.4, 64.3)	
Mismatch	37 (1.3) vs. 54 (2)	40.4 (9.4, 60.8)	
145a			
Match	183 (6.4) vs. 415 (15.1)	57.6 (49.6, 64.4)	
Mismatch	37 (1.3) vs. 53 (1.9)	39.3 (7.6, 60.1)	
152			
Match	216 (7.6) vs. 452 (16.4)	54.7 (46.7, 61.5)	
Mismatch	4 (0.1) vs. 16 (0.6)	75.9 (27.9, 91.9)	
190			
Match	147 (5.2) vs. 357 (13)	60.8 (52.5, 67.6)	
Mismatch	73 (2.6) vs. 111 (4)	38.4 (17.2, 54.2)	
346			
Match	180 (6.3) vs. 410 (14.9)	57.8 (49.7, 64.6)	
Mismatch	40 (1.4) vs. 58 (2.1)	39.9 (10.1, 59.8)	
414			
Match	216 (7.6) vs. 437 (15.9)	53.2 (44.9, 60.2)	
Mismatch	4 (0.1) vs. 31 (1.1)	87.5 (64.5, 95.6)	
417			
Match	147 (5.2) vs. 357 (13)	60.8 (52.5, 67.6)	
Mismatch	73 (2.6) vs. 111 (4)	38.4 (17.2, 54.2)	
484			
Match	67 (2.4) vs. 187 (6.8)	65.1 (53.9, 73.6)	
Mismatch	153 (5.4) vs. 281 (10.2)	49.1 (38.0, 58.2)	
501			
Match	104 (3.7) vs. 290 (10.5)	65.0 (56.2, 72.0)	
Mismatch	116 (4.1) vs. 178 (6.5)	40.5 (24.8, 52.9)	
655			
Match	146 (5.1) vs. 353 (12.8)	60.6 (52.2, 67.5)	
Mismatch	74 (2.6) vs. 115 (4.2)	39.7 (19.2, 54.9)	
681			
Match	176 (6.2) vs. 393 (14.3)	56.9 (48.5, 63.9)	
Mismatch	44 (1.5) vs. 75 (2.7)	48.4 (25.1, 64.4)	
778			
Match	219 (7.7) vs. 448 (16.3)	53.7 (45.5, 60.6)	
Mismatch	1 (0) vs. 20 (0.7)	95.1 (63.8, 99.3)	
950			
Match	180 (6.3) vs. 404 (14.7)	57.1 (48.9, 64.0)	
Mismatch	40 (1.4) vs. 64 (2.3)	45.5 (19.1, 63.3)	
1027			
Match	147 (5.2) vs. 357 (13)	60.8 (52.5, 67.6)	
Mismatch	73 (2.6) vs. 111 (4)	38.4 (17.2, 54.2)	
1167			
Match	207 (7.3) vs. 434 (15.8)	54.8 (46.7, 61.7)	
Mismatch	13 (0.5) vs. 34 (1.2)	63.0 (29.9, 80.5)	
1176			
Match	108 (3.8) vs. 239 (8.7)	57.6 (46.7, 66.2)	
Mismatch	112 (3.9) vs. 229 (8.3)	53.2 (41.3, 62.6)	

Table R2. A) Estimation and inference in the per-protocol cohort about VE against the primary (moderate to severe-critical) COVID-19 endpoint with match or mismatch to the vaccine insert at position 414, 501, and 778 in Latin America (i.e. a subset of the amino acid positions shown in Table R1). This analysis restricts cases to lineages that are not Lambda. This is Supplementary Fig. 50 in the revision. B) Comparison of the results for VE (%) (95% CI) for match vs. mismatch at positions 414, 501, and 778 for the all-lineage analysis and for the Lambda-removed analysis (results extracted from Fig. 2d and Table R2A).

A.

B.

Spike Amino Acid Position	VE (%) (95% CI) for Match vs. Mismatch at the Designated Position	
	All Lineages (Fig. 2d)	Lambda-Removed (Tables R1, R2)
414	49.8 (42.2, 56.4) vs. 86.5 (64.1, 94.9)	53.2 (44.9, 60.2) vs. 87.5 (64.5, 95.6)
501	58.0 (49.5, 65.1) vs. 40.9 (26.1, 52.8)	65.0 (56.2, 72.0) vs. 40.5 (24.8, 52.9)
778	50.3 (42.8, 56.8) vs. 94.1 (64.7, 99.0)	53.7 (45.5, 60.6) vs. 95.1 (63.8, 99.3)

Table R3. Estimation and inference in the per-protocol cohort about VE against the primary (moderate to severe-critical) COVID-19 endpoint with a specific screened-in amino acid residue. This analysis restricts cases to lineages that are not Lambda.

Geographic Region	Spike Amino Acid Position	Residue	Residue Present		Residue Absent	
			No. of Cases (V vs. P) (Incidence per 100 PYRs)	VE (%) (95% CI)	No. of Cases (V vs. P) (Incidence per 100 PYRs)	VE (%) (95% CI)
Latin America	18	F	76 (2.7) vs. 115 (4.2)	38.0 (17.1, 53.6)	144 (5.1) vs. 353 (12.8)	61.2 (52.8, 68.0)
	18	L	144 (5.1) vs. 353 (12.8)	61.2 (52.8, 68.0)	76 (2.7) vs. 115 (4.2)	38.0 (17.1, 53.6)
	20	N	72 (2.5) vs. 109 (4)	38.2 (16.7, 54.1)	148 (5.2) vs. 359 (13)	60.7 (52.4, 67.6)
	20	T	147 (5.2) vs. 359 (13)	61.0 (52.7, 67.8)	73 (2.6) vs. 109 (4)	37.3 (15.7, 53.4)
	26	P	147 (5.2) vs. 357 (13)	60.8 (52.5, 67.6)	73 (2.6) vs. 111 (4)	38.4 (17.2, 54.2)
	26	S	73 (2.6) vs. 111 (4)	38.4 (17.2, 54.2)	147 (5.2) vs. 357 (13)	60.8 (52.5, 67.6)
	69	gap	6 (0.2) vs. 17 (0.6)	68.0 (18.9, 87.4)	214 (7.5) vs. 451 (16.4)	54.9 (47.0, 61.7)
	69	H	214 (7.5) vs. 451 (16.4)	54.9 (47.0, 61.7)	6 (0.2) vs. 17 (0.6)	68.0 (18.9, 87.4)
	70	gap	6 (0.2) vs. 17 (0.6)	68.0 (18.9, 87.4)	214 (7.5) vs. 451 (16.4)	54.9 (47.0, 61.7)
	70	V	214 (7.5) vs. 451 (16.4)	54.9 (47.0, 61.7)	6 (0.2) vs. 17 (0.6)	68.0 (18.9, 87.4)
	95	I	43 (1.5) vs. 66 (2.4)	43.0 (16.3, 61.2)	177 (6.2) vs. 402 (14.6)	57.6 (49.4, 64.5)
	95	T	177 (6.2) vs. 401 (14.6)	57.5 (49.3, 64.4)	43 (1.5) vs. 67 (2.4)	43.8 (17.6, 61.7)
	138	D	145 (5.1) vs. 353 (12.8)	60.9 (52.5, 67.8)	75 (2.6) vs. 115 (4.2)	38.8 (18.2, 54.3)
	138	Y	74 (2.6) vs. 114 (4.1)	39.1 (18.5, 54.6)	146 (5.1) vs. 354 (12.9)	60.7 (52.4, 67.6)
	144	gap	12 (0.4) vs. 27 (1)	59.0 (19.1, 79.2)	208 (7.3) vs. 441 (16)	55.2 (47.2, 62.0)
	144	T	34 (1.2) vs. 52 (1.9)	43.3 (12.7, 63.2)	186 (6.5) vs. 416 (15.1)	57.0 (48.9, 63.9)
	144	Y	170 (6) vs. 385 (14)	57.4 (49.0, 64.4)	50 (1.8) vs. 83 (3)	46.7 (24.3, 62.5)
	145	S	32 (1.1) vs. 49 (1.8)	43.3 (11.5, 63.7)	188 (6.6) vs. 419 (15.2)	56.9 (48.8, 63.7)
	145	Y	183 (6.4) vs. 414 (15)	57.5 (49.4, 64.3)	37 (1.3) vs. 54 (2)	40.4 (9.4, 60.8)
	145a	gap	183 (6.4) vs. 415 (15.1)	57.6 (49.6, 64.4)	37 (1.3) vs. 53 (1.9)	39.3 (7.6, 60.1)
	145a	N	37 (1.3) vs. 53 (1.9)	39.3 (7.6, 60.1)	183 (6.4) vs. 415 (15.1)	57.6 (49.6, 64.4)
	152	W	216 (7.6) vs. 452 (16.4)	54.7 (46.7, 61.5)	4 (0.1) vs. 16 (0.6)	75.9 (27.9, 91.9)
	190	R	147 (5.2) vs. 357 (13)	60.8 (52.5, 67.6)	73 (2.6) vs. 111 (4)	38.4 (17.2, 54.2)
	190	S	73 (2.6) vs. 111 (4)	38.4 (17.2, 54.2)	147 (5.2) vs. 357 (13)	60.8 (52.5, 67.6)
	346	K	40 (1.4) vs. 58 (2.1)	39.9 (10.1, 59.8)	180 (6.3) vs. 410 (14.9)	57.8 (49.7, 64.6)
	346	R	180 (6.3) vs. 410 (14.9)	57.8 (49.7, 64.6)	40 (1.4) vs. 58 (2.1)	39.9 (10.1, 59.8)
	414	Q	216 (7.6) vs. 437 (15.9)	53.2 (44.9, 60.2)	4 (0.1) vs. 31 (1.1)	87.5 (64.5, 95.6)
	414	R	4 (0.1) vs. 31 (1.1)	87.5 (64.5, 95.6)	216 (7.6) vs. 437 (15.9)	53.2 (44.9, 60.2)
	417	K	147 (5.2) vs. 357 (13)	60.8 (52.5, 67.6)	73 (2.6) vs. 111 (4)	38.4 (17.2, 54.2)
	417	T	73 (2.6) vs. 111 (4)	38.4 (17.2, 54.2)	147 (5.2) vs. 357 (13)	60.8 (52.5, 67.6)
	484	E	67 (2.4) vs. 187 (6.8)	65.1 (53.9, 73.6)	153 (5.4) vs. 281 (10.2)	49.1 (38.0, 58.2)
	484	K	153 (5.4) vs. 281 (10.2)	49.1 (38.0, 58.2)	67 (2.4) vs. 187 (6.8)	65.1 (53.9, 73.6)
	501	N	104 (3.7) vs. 290 (10.5)	65.0 (56.2, 72.0)	116 (4.1) vs. 178 (6.5)	40.5 (24.8, 52.9)
	501	Y	116 (4.1) vs. 178 (6.5)	40.5 (24.8, 52.9)	104 (3.7) vs. 290 (10.5)	65.0 (56.2, 72.0)
	655	H	146 (5.1) vs. 353 (12.8)	60.6 (52.2, 67.5)	74 (2.6) vs. 115 (4.2)	39.7 (19.2, 54.9)
	655	Y	74 (2.6) vs. 115 (4.2)	39.7 (19.2, 54.9)	146 (5.1) vs. 353 (12.8)	60.6 (52.2, 67.5)
	681	H	44 (1.5) vs. 74 (2.7)	47.7 (24.0, 64.0)	176 (6.2) vs. 394 (14.3)	57.0 (48.6, 64.0)
	681	P	176 (6.2) vs. 393 (14.3)	56.9 (48.5, 63.9)	44 (1.5) vs. 75 (2.7)	48.4 (25.1, 64.4)
	778	I	1 (0) vs. 20 (0.7)	95.1 (63.8, 99.3)	219 (7.7) vs. 448 (16.3)	53.7 (45.5, 60.6)
	778	T	219 (7.7) vs. 448 (16.3)	53.7 (45.5, 60.6)	1 (0) vs. 20 (0.7)	95.1 (63.8, 99.3)
	950	D	180 (6.3) vs. 404 (14.7)	57.1 (48.9, 64.0)	40 (1.4) vs. 64 (2.3)	45.5 (19.1, 63.3)
	950	N	40 (1.4) vs. 64 (2.3)	45.5 (19.1, 63.3)	180 (6.3) vs. 404 (14.7)	57.1 (48.9, 64.0)
	1027	I	73 (2.6) vs. 111 (4)	38.4 (17.2, 54.2)	147 (5.2) vs. 357 (13)	60.8 (52.5, 67.6)
	1027	T	147 (5.2) vs. 357 (13)	60.8 (52.5, 67.6)	73 (2.6) vs. 111 (4)	38.4 (17.2, 54.2)
	1167	A	13 (0.5) vs. 34 (1.2)	63.0 (29.9, 80.5)	207 (7.3) vs. 434 (15.8)	54.8 (46.7, 61.7)
	1167	G	207 (7.3) vs. 434 (15.8)	54.8 (46.7, 61.7)	13 (0.5) vs. 34 (1.2)	63.0 (29.9, 80.5)
	1176	F	112 (3.9) vs. 229 (8.3)	53.2 (41.3, 62.6)	108 (3.8) vs. 239 (8.7)	57.6 (46.7, 66.2)
	1176	V	108 (3.8) vs. 239 (8.7)	57.6 (46.7, 66.2)	112 (3.9) vs. 229 (8.3)	53.2 (41.3, 62.6)

In addition, we thank the reviewer for the helpful point about phylogenetic correction and acknowledge that not accounting for phylogenetic relationships can lead to associations of AA changes with VE that are due to bystander mutations within distinct lineage. We conducted the GenSig analysis (the LANL tool designed by Korber et al.) for the 37 screened-in sites evaluated elsewhere in the manuscript for Latin American data, and the results are reported in the new subsection of Results. While this approach confirms some (but not all) of the signature sites previously found to be associated with VE, we note that this method has its own limitations. GenSig was specifically designed to analyze HIV-1 sequences, which are far more diverse than SARS-CoV-2 sequences. And because GenSig scans the full protein length, with less diversity, it requires a vast

number of sequences to be sufficiently powered to detect true associations. Additionally, because phylogenetic reconstructions are built on a nucleotide alignment, and part of our data set was made of imputed amino acid Spike regions, we ended up with a smaller set of sequences available for the GenSig input dataset. Nonetheless, we were able to confirm the significant association of VE with the Lambda variant deletion at positions 248-252 in the NTD supersite epitope, and this association withstood phylogenetic correction. The new Supplementary Table 25 reports the results. We agree with the reviewer's observation that while we are likely still underpowered to identify the true mutations that drive the reduced VE within the Lambda spike sequences, these analyses confirm that the Lambda variant is indeed more likely to escape vaccine-induced responses. This is in line with findings that the Lambda variant exhibited higher infectivity and resistance to cellular immune responses as well as to neutralizing antibodies.^{9,10}

We also performed an additional literature search, and now elaborated the discussion to cite and discuss additional work related to the Lambda variant, the sieve sites, and their association with antibody neutralization, as well as their prevalence in current circulating strains (this new discussion is added to multiple paragraphs starting at the fourth paragraph of the Discussion). These findings shed light on which sieve sites appear to be evolutionarily important, and they provide some interpretations for the discordancy of the ordering of variants by the level of neutralization resistance to Ad26.COVS vaccinee sera compared to the ordering of the level of vaccine efficacy against the variants. We thank the reviewer for raising the issue of this discordance, as it is an important issue to discuss in evaluating the role of neutralizing antibodies as a correlate of protection.

The following issues will help improve readability of this manuscript:

1) Figure S4 is quite telling and I would urge the authors to include it as a panel in a main figure.

Response: This is now panel (e) of Fig. 2.

2) Lines 150-3: It is not clear if the results for the 38 residues are shown in any supplementary item. If not, please consider displaying these results.

Response: These 38 residues/residue features are shown in the logo plot that is now Fig. 2e. We have revised the sentence as follows: "Similarly, when assessing the presence or absence of specific residues at each AA position, VE significantly differed (q-value \leq 0.20) for 38 residues/residue features (75V vs. not-75V and 76I vs. not-76I with FWER $p \leq$ 0.05) distributed across these 16 positions (Fig. 2e)."

3) Lines 189-221: Current phrasing is unclear if the DMS studies were done in this work, or previously. The supplementary information does not describe how the different DMS maps were obtained. I am assuming that these are from different NAb sera, but it would help to add a column to Table S1 indicating which NAb sera gave rise to each DMS cluster.

It will help readers to be able to connect the DMS/PDB clusters to known NAb classification schemes such as Barnes et al. Class 1-4 (<https://pubmed.ncbi.nlm.nih.gov/33045718/>) or Hastie et al. (<https://pubmed.ncbi.nlm.nih.gov/34554826/>) and/or giving examples of antibodies that typify

interesting clusters. Because such antibody classes or monoclonals will be more informative to readers, I would also recommend using these clusters or specific monoclonal antibody names in the nomenclature of the DMS cluster, for example "DMS1 (Barnes et al. Class XXX)" or "DMS2 (monoclonal XXX derived)", etc.

Response: The DMS studies were done previously in Greaney et al.¹¹ The following text has been added to the section *Deep mutational scanning (DMS) antibody escape scores* to better describe how we derived.

Based on deep mutational scanning data from 228 SARS-CoV-2-elicited neutralizing antibodies (data captured April 21, 2022; https://raw.githubusercontent.com/jbloombloom/SARS2_RBD_Ab_escape_maps/651fe6fa5a7fccec2b662d4bb45b6d2c7421ae74/processed_data/escape_calculator_data.csv),¹¹ we used hierarchical clustering based on Euclidean distances and complete linkage for the purpose of identifying potential antibody footprints that would be defined by a set of amino acid sites. Based on the goal of identifying approximately 15 putative epitope footprint sites to define a cluster and a relatively small number of clusters, we cut the resulting dendrogram at heights that generated between K=2 and 21 non-singleton clusters from the tree. Singleton clusters were deemed to be uninformative and were dropped from further analysis. For each cluster we selected sites with an average escape of greater than 0.05 to define a putative antibody footprint for the cluster. Based on these criteria we selected K=8 non-singleton clusters whose defining putative epitope footprint amino acid sites are shown in Table S1.

Additional work by Greaney et al.¹² maps three of the four Barnes et al. classes to specific sites in the RBD. We have added annotation to Table S1 to show the number of sites in our putative epitope footprints that overlap with these Class 1, 2 and 3 sites.

We also considered the Hastie et al. paper, but the antibodies used in that study do not overlap with the antibodies used in Greaney et al.¹¹

5) Lines 225-9: The motivation and description of these NTD features is buried in the supplement and SAP. Please consider briefly mentioning these in the main text, and using descriptive names instead of "NTD1"- "NTD7" (e.g. "Indel 138-158/242-264" for NTD1, or "Mutation 12-13" for NTD2) such that readers can identify the Spike sites.

Response: We have moved the entire section "Neutralization hypothesis-driven sieve analysis" from the Supplementary Materials and Methods to the Methods section in the main text.

6) Lines 245-248: Please consider briefly summarizing results from the variable importance analyses in the main text.

Response: We have added the following results to the main text:

"A variable importance measure (VIM) analysis by ensemble machine learning¹³ was conducted to assess how well different groups of virus features in COVID-19 endpoint cases predicted treatment

arm beyond that provided by baseline risk factors (whether the COVID-19 endpoint was from Colombia and enrollment periods). Virus features defined based on neutralization data were the top-performing predictors of treatment arm, with the DMS RBD antibody-escape features having the highest estimated VIM (0.073, p-value for a test of the null hypothesis of zero VIM = 0.043; Supplementary Fig. 42).

The second-most important classifying variables were the set of all nAb CoP hypothesis features, defined as DMS RBD and PDB antibody-escape features, NTD neutralization-relevant features, and the variant-neutralization sensitivity score (VIM = 0.051); PDB antibody-escape features (VIM = 0.049); and the variant-neutralization sensitivity score (VIM = 0.048) (Supplementary Fig. 42). The “unbiased” features specified to include all Spike AA variation ignoring neutralization hypotheses had the lowest estimated variable importance (VIM = 0.036 to 0.046 for the weighted Hamming distances and Spike AAs).”

7) Discussion: It is clear that Lambda drives the sequence features impacting VE -- the limitation that there could be bystander mutations in the set of mutations identified is not discussed. The interesting finding that perhaps Lambda neutralization resistance might not capture the primary endpoint VE for Lambda is not discussed.

Response: We now discuss the potential of Lambda’s “bystander” mutations in the Discussion (pp. 20-21), and on pp. 21-22 we also discuss the possibility that Lambda might show increased resistance to other immune responses besides neutralization, specifically cellular responses. However, we stress there is tenuous evidence at best to support such a hypothesis.

References

1. Bergeri I, Whelan MG, Ware H, et al. Global SARS-CoV-2 seroprevalence from January 2020 to April 2022: A systematic review and meta-analysis of standardized population-based studies. *PLoS Med* 2022; **19**(11): e1004107.
2. Jones JM, Manrique IM, Stone MS, et al. Estimates of SARS-CoV-2 Seroprevalence and Incidence of Primary SARS-CoV-2 Infections Among Blood Donors, by COVID-19 Vaccination Status — United States, April 2021–September 2022. *MMWR Morb Mortal Wkly Rep* 2023; **72**: 601-5.
3. Bingham J, Cable R, Coleman C, et al. Estimates of prevalence of anti-SARS-CoV-2 antibodies among blood donors in South Africa in March 2022. *Res Sq* 2022.
4. Madhi SA, Kwatra G, Myers JE, et al. Population Immunity and Covid-19 Severity with Omicron Variant in South Africa. *N Engl J Med* 2022; **386**(14): 1314-26.
5. Sykes W, Mhlanga L, Swanevelder R, et al. Prevalence of anti-SARS-CoV-2 antibodies among blood donors in Northern Cape, KwaZulu-Natal, Eastern Cape, and Free State provinces of South Africa in January 2021. *Res Sq* 2021.
6. Sadoff J, Gray G, Vandebosch A, et al. Safety and Efficacy of Single-Dose Ad26.COVS.2.S Vaccine against Covid-19. *N Engl J Med* 2021; **384**(23): 2187-201.
7. Sadoff J, Gray G, Vandebosch A, et al. Final Analysis of Efficacy and Safety of Single-Dose Ad26.COVS.2.S. *N Engl J Med* 2022; **386**(9): 847-60.
8. Moyo-Gwete T, Richardson SI, Keeton R, et al. Homologous Ad26.COVS.2.S vaccination results in reduced boosting of humoral responses in hybrid immunity, but elicits antibodies of similar magnitude regardless of prior infection. *medRxiv* 2023.
9. Kimura I, Kosugi Y, Wu J, et al. The SARS-CoV-2 Lambda variant exhibits enhanced infectivity and immune resistance. *Cell Rep* 2022; **38**(2): 110218.

10. Motozono C, Toyoda M, Zahradnik J, et al. SARS-CoV-2 spike L452R variant evades cellular immunity and increases infectivity. *Cell Host Microbe* 2021; **29**(7): 1124-36 e11.
11. Greaney AJ, Starr TN, Bloom JD. An antibody-escape estimator for mutations to the SARS-CoV-2 receptor-binding domain. *Virus Evol* 2022; **8**(1): veac021.
12. Greaney AJ, Starr TN, Barnes CO, et al. Mapping mutations to the SARS-CoV-2 RBD that escape binding by different classes of antibodies. *Nat Commun* 2021; **12**(1): 4196.
13. Williamson B, Gilbert PB, Simon N, Carone M. A General Framework for Inference on Algorithm-Agnostic Variable Importance. doi: 10.1080/01621459.2021.2003200. *Journal of the American Statistical Association* 2022.

Reviewers' Comments:

Reviewer #1:

Remarks to the Author:

I am happy with the revision and believe the paper is now ready for publication.

Reviewer #2:

Remarks to the Author:

Authors have addressed all my comments satisfactorily and I'm happy to endorse this manuscript .

Reviewer #3:

None